# Neighborhood-Informed Diffusion Model for Source-Free Domain Adaptation: Retrieving Source Ground Truth from Target Query's Neighbors

## Abstract

Diffusion models, empowered as an input augmentation technique, have demonstrated promise in domain adaptation. However, to effectively capture shared characteristics between two data densities, such a diffusion model needs to be trained using both source and target data for its generation. This constraint narrows its application to a more demanding yet authentic scenario where source data remains inaccessible during target adaptation, *i.e.,* source-free domain adaptation (SFDA). In the absence of source data during adaptation, which hinders the analytical quantification of domain shift, can we employ the pre-trained source representation to formulate a diffusion model for facilitating the unsupervised clustering in target adaptation? To answer this question, we introduce a novel method, *discriminative neighborhood diffusion* (DND). DND transforms the pre-trained source representation into a target-to-source diffusion model by parameterizing the prior densities of the diffusion process, leveraging the smoothness indicated by latent $k$-nearest neighbors ($k$-NNs). The samples generated from the diffusion model are then used as positive keys for contrastive clustering during adaptation. This process effectively introduces a form of supervision into unsupervised clustering by incorporating the latent geometries from both the source and target domains' latent $k$-NNs. By evaluating DND against various SFDA methods on multiple benchmark datasets, we demonstrate the discriminative potential of diffusion models in the absence of source data. Moreover, the effectiveness of DND is demonstrated as it successfully solves SFDA problems, achieving state-of-the-art performance.

## 1 Introduction

How can we generate informative features when we have limited annotated data points? This challenge intersects the fields of semi-supervised learning (SSL) and domain adaptation (DA). SSL relies on the smoothness assumption, which posits that neighboring data points are likely to share similar labels, thus facilitating the propagation of knowledge from labeled to nearby unlabeled points (Iscen et al., 2019). In contrast, DA focuses on counteracting the decline in model performance due to shifts in data distributions, by learning shared representations from diverse data distributions (Ben-David et al., 2006), and is known to enhance the generalizability of models (Radford et al., 2018). In this study, drawing inspiration from SSL and DA, we employ a generative model, specifically a diffusion model, to address source-free domain adaptation (SFDA) problems. SFDA primarily involves adapting models to new target domains without using source data during target adaptation. Our method centers on the SSL principle of latent geometry, discerned through the k-nearest neighbors (k-NNs) in latent space, which explicitly guides the knowledge transfer for SFDA. To implement this, we use generative models to transform target latent features, imparting them with source-specific traits, under the guidance of latent geometry in the absence of source data.

In brief, we aim to explore how generative models can acquire and utilize their discriminative potential by leveraging the concept of *smoothness*. Our research investigates how a generative model, with a prior distribution parameterized by the latent geometry of a specific (*source*) data density, can disperse the ground truth knowledge of its data points within their respective neighborhoods in

latent space during training. Subsequently, this model can retrieve this knowledge near data points that exhibit similar latent geometry within a new, similar (*target*) data density. This strategy enables the seamless use of the pre-trained source classifier in the target domain. It eliminates the need for both access to source data and updates to the classifier's model parameters during target adaptation. Our diffusion model stands apart from typical generative models, which aim to generate new data points. Instead, we employ this model as a *domain shift corrector*. Its purpose is to refine the features of the unlabeled target domain using knowledge learned from the labeled source domain, all without needing to access source data. More importantly, the smoothness in latent geometry guarantees that there is a match between the transformed target features and the source-specific ground truth knowledge retrieved from our diffusion model. This approach successfully addresses the most challenging aspect of SFDA, which is to explicitly transfer knowledge without access to source data.

We name this method *discriminative neighborhood diffusion* (DND). DND employs diffusion models, not for the purpose of image generation, but to refine feature spaces and guide clustering in SFDA. In general, diffusion models operate on thermodynamic principles, simplifying and reconstructing data densities in a manner similar to substance dispersion (Sohl-Dickstein et al., 2015). This similarity emphasizes the need for properly setting the prior in the diffusion process. Crucially, our approach involves aptly determining the Gaussian prior's mean and variance within diffusion. Our diffusion process, by tuning parameters in accordance with latent geometry during both source pre-training and target adaptation phases, refines target features to match source-specific characteristics. These characteristics relate to specific source ground truths and adapt to changing data densities, effectively addressing domain shifts.

Our approach, depicted in Figure 1, is contextualized within SFDA: **Source Pre-training**: We start by training a classifier on a labeled source dataset to create a feature space and decision boundary. Concurrently, we train a diffusion model to generate features beyond the query data point's encoded space, focusing on its neighborhood. This process ensures that the features generated closely match the query's ground truth. The diffusion model is specifically trained to produce features that stay close to the ground truth class yet away from the decision boundary, as shown in Figure 1. **Target Adaptation**: During this phase, we keep the diffusion model and classifier parameters fixed, but update the encoder. The aim is to extract the source ground truth from each target query's diffusion profile and realign the target features to the nearby source decision boundary. This phase is essential for successful unsupervised clustering, as it utilizes explicit guidance from supervised pre-training. It effectively transforms target features to resemble those of the source, thereby steering them towards the proximate source decision boundary. Our contributions are threefold:

- We introduce a diffusion model with a discriminative focus for SFDA, validated by experiments on classification benchmarks to assess domain shift (see Appendices).

- We incorporate latent geometry, derived from $k$-NNs, into our model. This integration facilitates the storage and retrieval of ground truth knowledge across various data distributions, ensuring the correspondence between the encoded target features and the source knowledge retrieved.

- Our method demonstrates superior performance in SFDA, showcasing its ability to transfer knowledge without source data.

## 2 RELATED WORK

**Contrastive Source-Free Domain Adaptation**  SFDA is a transfer learning task of adapting a model trained on one labeled data domain to perform well on another unlabeled domain with different data densities. Compared with other DA settings, this adaptation lacks access to source data, adding complexity to domain shift estimation (Liang et al., 2020). Contrastive learning, a powerful SSL technique, enhances model robustness by pre-training on unsupervised clustering in the latent space (Chen et al., 2020). In SFDA, the tasks in both domains are identical, with the same number of classes, which allows us to formulate the contrastive objective in the output probability space, facilitating direct clustering of samples based on their task-specific classes. The superior performance of contrastive SFDA (Yang et al., 2022; Zhang et al., 2022), as demonstrated in comparison to other methods, strongly motivates us to explore the discriminative essence of diffusion generative models. Several compelling reasons support our choice to build our research upon contrastive SFDA:

- **Assessing the discriminativeness of diffusion-generated features.** Transferring knowledge from labeled to unlabeled data helps assess the discriminative quality of the generated features.

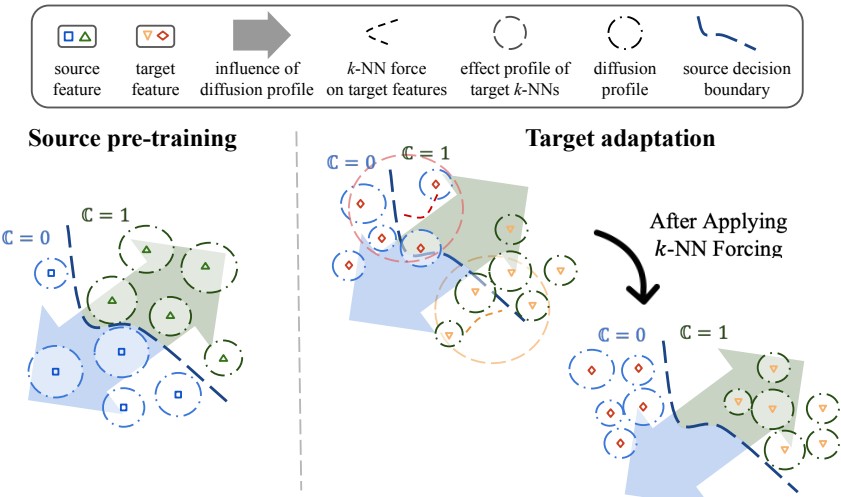

Figure 1: **Motivation.** We simplify the problem into binary classification, where squares and diamonds belong to class **0**, and triangles and inverted triangles belong to class **1**. Squares and triangles originate from source domain, while diamonds and inverted triangles are from target domain. The diffusion profile size is determined by on the mean and variance of the query's $k$-NN features.

- **Separate supervised source and unsupervised target training.** The training is split into source pre-training and unsupervised target adaptation, helping us capture the level of supervision from the generated features by evaluating the performance in classifying unlabeled target samples.

- **Contrastive clustering.** The accuracy of the clusters formed via contrastive learning largely depends on the positive keys, allowing us to assess the discriminative essence of diffusion models.

**Diffusion models** Diffusion generative models (DGMs) are categorized into stochastic and deterministic types based on noise injection into stochastic differential equations (SDEs) and their use of ordinary differential equations (ODEs) to nullify this noise. The standard approach to formulate DGMs involves representing each diffusion step as the solution to an SDE (Ho et al., 2020). This SDE accounts for random fluctuations due to Brownian motion, while drifts in the SDE indicate the deterministic paths of the diffusion process, modeled using ODEs (for further details on formulating the diffusion step using SDE, please refer to Appendix B). We aim to demonstrate the potential of diffusion models in enhancing discriminative processes rather than performance improvement. To simplify, we avoid complex covariance matrix estimation related to Brownian motion (Song et al., 2021) and adopt a deterministic diffusion model with basic sampling strategies (Heitz et al., 2023). In our work, each diffusion step is solely represented by the deterministic path of a noisy sample projected onto the data density.

**Domain adaptation using diffusion models** Recently, DGMs have gained attention as an input augmentation technique in the domain adaptation community. For instance, a text-to-image diffusion model was used to synthesize the target domain with source labels, showcasing the effectiveness of diffusion models in one-shot unsupervised DA (Benigmim et al., 2023). This text-guided domain adaptation technique has also proven effective in transferring knowledge from 3D generative models (Kim & Chun, 2023). Moreover, DGMs learned from multiple source domains have been used to condition approximate inference on the target domain (Graikos et al., 2022). However, existing research has focused on augmenting training data using DGMs without considering incorporating tasks that are inherently discriminative. Furthermore, none of the existing studies has demonstrated the effectiveness of DGMs in transferring knowledge for SFDA.

**Source-free domain adaptation using generative models** In the absence of source data during target adaptation, using generative models to generate source-like images or feature prototypes is a typical strategy for SFDA. CPGA (Qiu et al., 2021) generates class-specific avatar feature prototypes via contrastive learning, aligning pseudo-labeled target features for domain adaptation without source data. 3C-GAN (Li et al., 2020) combines a discriminator, generator for target-style samples,

and a pre-trained predictor to enhance target domain accuracy, but requires careful regularization to balance classification and image generation training objectives. Differently, our method employs neighborhood-informed diffusion models to transform target features into source-like ones, serving as a *domain shift corrector*. This approach, using the diffusion model's prior density for feature transformation, represents a significant departure from conventional domain adaptation strategies.

## 3 METHOD

### 3.1 PRELIMINARY

**Notations** We denote a set of data points from the source domain as $\mathbf{S} = \{(\mathbf{x}^{s,i}, \mathbf{y}^{s,i})\}_{i=1}^{N}$ where $\mathbf{x}^{s,i}$ represents $i$-th source input, and $\mathbf{y}^{s,i}$ denotes its corresponding ground truth. Also, a set of data points drawn from the target domain is denoted as $\mathbf{T} = \{\mathbf{x}^{t,i}\}_{i=1}^{M}$ with $\mathbf{x}^{t,i}$ denoting $i$-th target input. Here, no access to target labels $\mathbf{y}^t$ is available during adaptation. A classification model $f$, which predicts the ground truth for both data domains, consists of an encoder $G$ and a classifier $F$ *i.e.* $f = F \circ G$. The deterministic paths of the diffusion model $D$ are also parameterized using neural nets.

$G$ operates by converting input data into their latent features (*i.e.,* encoded features). $F$ converts the encoded or generated features (from $D$) into probability vectors, with each dimension corresponding to a one-hot encoding of a class. Meanwhile, $D$ employs prior samples, matching the dimensionality of encoded features, to generate discriminative features that extend beyond the encoded space.

**Neighborhood definition** The *neighborhood* of a query data is defined by its $k$-NNs in the encoded space, where we measure similarity within a feature bank that includes features encoded from all training data. To elaborate, we use the feature bank established in the source domain as an example: $\mathbb{B}_{\mathbf{S}} := \{G_s(\mathbf{x}^{s,j}) \mid \mathbf{x}^{s,j} \in \mathbf{S}\}_{j=1}^{N}$, where $G_s$ is the encoder optimized using source samples. Similar to (Huang et al., 2019), given a query's feature vector $\mathbf{z}^{s,i} = G_s(\mathbf{x}^{s,i})$, we employ the cosine similarity to determine its $k$-NNs within $\mathbb{B}_{\mathbf{S}}$:

$$d(\mathbf{z}^{s,i}, \mathbb{B}_{\mathbf{S}}^{j}) := \frac{\mathbf{z}^{s,i} \cdot \mathbb{B}_{\mathbf{S}}^{j}}{||\mathbf{z}^{s,i}|| \cdot ||\mathbb{B}_{\mathbf{S}}^{j}||}, \tag{1}$$

where $\mathbb{B}_{\mathbf{S}}^{j}$ denotes the $j$-th element of $\mathbb{B}_{\mathbf{S}}$ that corresponds to an input data $\mathbf{x}^{s,j}$. We use different notations for the query's encoded features $\mathbf{z}^{s,i}$ and the features $\mathbb{B}_{\mathbf{S}}^{j}$ stored in the feature bank due to potential differences before updating the feature bank with the current encoder parameters. Therefore, the neighborhood comprising $k$-NNs of the query $\mathbf{z}^{s,i}$ is defined as:

$$\mathbb{N}_k(\mathbf{z}^{s,i}) := \arg min_{max(\mathcal{K}) \leq N, |\mathcal{K}| = k} \sum_{j \in \mathcal{K}} d(\mathbf{z}^{s,i}, \mathbb{B}_{\mathbf{S}}^{j}), \tag{2}$$

where $\mathcal{K}$ is a set that includes the indices of all $k$-NNs in $\mathbb{B}_{\mathbf{S}}$.

**Diffusion model formulation** To transition from a stable density sample $\mathbf{z}_0$ to a complex density sample $\mathbf{z}_1$, we use the deterministic diffusion model known as Iterative $\alpha$-(De)blending (IADB) (Heitz et al., 2023). This model, inspired by blending and deblending operations in image editing, transforms $(\mathbf{z}_0, \mathbf{z}_1)$ into $\mathbf{z}_\alpha$ using a blending coefficient $\alpha$, and vice versa. Although the posterior densities from deblending, $(\hat{\mathbf{z}}_0, \hat{\mathbf{z}}_1)$, may differ from the initial densities $(p_0 \times p_1)$, the law of total probability allows us to revert these posterior densities to the initial ones by summing over $\mathbf{z}_\alpha$ sampled from $p_\alpha$. Essentially, the stochastic mapping between two latent densities, $p_{\alpha_1}$ and $p_{\alpha_2}$, can be simplified into two stages: $\alpha_1$-deblending and $\alpha_2$-blending, as expressed in Equation 3:

$$\mathbf{z}_{\alpha_1} \rightarrow (\mathbf{z}_0, \mathbf{z}_1) \rightarrow \mathbf{z}_{\alpha_2}, \tag{3}$$

where $\mathbf{z}_{\alpha_1} \sim p_{\alpha_1}$ and $\mathbf{z}_{\alpha_2} \sim p_{\alpha_2}$. This sampling is applied iteratively with blending parameters $\alpha_t = t/T$, where $t = 0, 1, ..., T$, to facilitate the transformation of $z_0 \sim p_0$ into $z_1 \sim p_1$ (refer to Algorithm 1). To ensure stability in this process, the expected values of posterior samples are calculated, thereby making it deterministic. A neural network is used to estimate the average difference between the posterior samples $\hat{\mathbf{z}}_1$ and $\hat{\mathbf{z}}_0$ at each $\alpha_t$. The training of $D$ is to align this estimated average difference with that of the initial densities (as detailed in Algorithm 2).

---

**Algorithm 1:** Sampling from Diffusion Models

---

1 **Input**: samples from initial densities $(\mathbf{z}_0, \mathbf{z}_1) \sim (p_0 \times p_1)$, time steps $T$, and $\alpha_t = \frac{t}{T}$;
2 **for** $t = 0, 1, ..., T-1$ **do**
3 $\quad \mathbf{z}_{\alpha_{t+1}} = \mathbf{z}_{\alpha_t} + (\alpha_{t+1} - \alpha_t) D(\mathbf{z}_{\alpha_t}, \alpha_t)$ ;
4 **end**
5 **Output**: posterior sample $\hat{\mathbf{z}}_1 = \mathbf{z}_{\alpha_T} \approx \mathbf{z}_1$.

---

**Algorithm 2:** Training Diffusion Models

---

1 **Input**: samples from initial densities $(\mathbf{z}_0, \mathbf{z}_1) \sim (p_0 \times p_1)$, blending parameters $\alpha \sim$ Uniform$[0, 1]$, and parameters $\phi$ for $D$;
2 **for** $t = 0, 1, ..., T$ **do**
3 $\quad \mathbf{z}_{\alpha_t} = (1 - \alpha_t)\mathbf{z}_0 + \alpha_t \mathbf{z}_1$ ;
4 $\quad$ update $\phi$ to minimize $\mathcal{L}_{dif} = \mathbb{E}_{\alpha_t, \mathbf{z}_{\alpha_t}}[||D_\phi(\mathbf{z}_{\alpha_t}, \alpha_t) - \mathbb{E}_{(\mathbf{z}_0, \mathbf{z}_1)|(\mathbf{z}_{\alpha_t}, \alpha_t)}[\mathbf{z}_1 - \mathbf{z}_0]||^2]$;
5 $\quad$ sample $\hat{\mathbf{z}}_1$ using **Algorithm 1** and update $\phi$ to minimize $\mathcal{L}_{ce} = \frac{1}{N} \sum_{i=1}^{N} \mathbb{1}[F_s(\hat{\mathbf{z}}_1) \neq \mathbf{y}^{s,i}]$;
6 **end**
7 **Output**: updated diffusion model parameters $\phi$.

---

### 3.2 DISCRIMINATIVE NEIGHBORHOOD DIFFUSION

In this section, we present our method *discriminative neighborhood diffusion* (DND), designed to tackle the SFDA problem, which operates in two phases: source pre-training and target adaptation. Notably, we highlight how DND **explicitly** transfers knowledge, distinguishing it from existing methods that do so **implicitly**.

#### 3.2.1 SOURCE PRE-TRAINING

**Source representation learning** The goal of this stage is to train the classification model $f_s$ using source data and their labels, with the training objective to minimize the source classification error: $\epsilon_{\mathbf{S}}(f_s) = \frac{1}{N} \sum_{i=1}^{N} \mathbb{1}[f_s(\mathbf{x}^{s,i}) \neq \mathbf{y}^{s,i}]$. We use the cross-entropy loss for optimizing the model parameters. This stage ensures that, as the training objective converges, the model's output probabilities closely match their corresponding ground truth.

**Diffusion model learning** The pre-trained source encoder $G_s$ and classifier $F$ are then used to generate samples for the initial densities $(\mathbf{z}_0, \mathbf{z}_1) \sim (p_0 \times p_1)$ for training the diffusion model, as outlined in Algorithm 2. To prevent confusion, we use the notation $(\mathbf{z}_0^s, \mathbf{z}_1^s) \sim (p_0^s \times p_1^s)$ to denote the initial densities in this stage, where $p_0^s$ is a Gaussian density parameterized by the latent $k$-NNs of $\mathbf{z}^s$. To be specific, we calculate the mean and variance of $p_0^s$ as follows: $\mu_0^s = \frac{1}{k} \sum_{j=1}^{k} \mathbb{N}_k^j(\mathbf{z}^s)$ and $\sigma_0^{s2} = \frac{1}{k} \sum_{j=1}^{k} (\mathbb{N}_k^j(\mathbf{z}^s) - \mu_0^s)^2$, which leads to $p_0^s \sim \mathcal{N}(\mu_0^s, \sigma_0^{s2})$. Moreover, $\mathbf{z}^s \sim p_1^s$ is the encoded feature density whose samples should be correctly classified to align with the source ground truth, hence the use of $F$.

During this stage, the parameters of both $G_s$ and $F$ remain fixed, with only the parameters of $D$ being updated. As illustrated in Figure 2, in each iteration, a mini-batch of $\mathbf{x}^s$ is fed into $G_s$, which transforms them into the corresponding $\mathbf{z}^s$. These features can be correctly classified into $\mathbf{y}^s$ by $F$. Subsequently, $\mathbf{z}^s$ are used to identify their latent $k$-NNs for the parameterization of the prior density $p_0^s$. As training progresses, $D$ should become capable of estimating the average difference between the **random** samples $\mathbf{z}_0^s$ and the **discriminative** features $\mathbf{z}^s$ used for classification. As a result, the ground truth information of $\mathbf{z}^s$ gradually spreads, within the latent feature space, to its neighbors.

#### 3.2.2 TARGET ADAPTATION

During target adaptation, the goal is to develop a target classification model $f_t$ that can minimize the generalization error in the target domain: $\epsilon_{\mathbf{T}}(f_t) = \frac{1}{M} \sum_{i=1}^{M} \mathbb{1}[f_t(\mathbf{x}^{t,i}) \neq \mathbf{y}^{t,i}]$ without relying on any $\mathbf{y}^t$. In this stage, we solely update $G_t$ while keeping the parameters of both $F$ and $D$ fixed. This

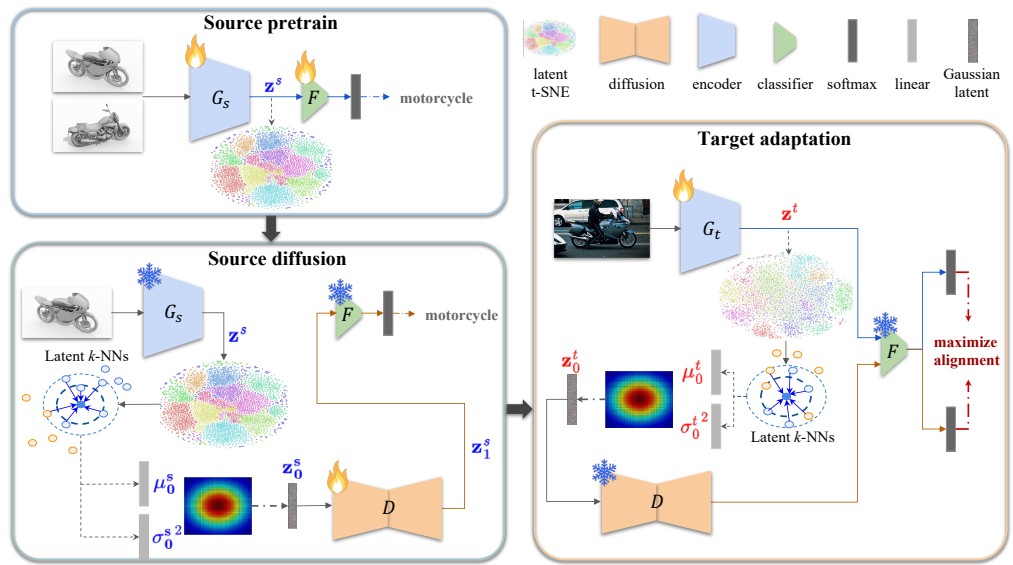

Figure 2: **Framework overview.** During training, the diffusion model gradually spreads ground truth knowledge within the neighborhoods of each source data point. Subsequently, during sampling, this knowledge is retrieved from the neighborhoods of target queries. The key for such knowledge storage and retrieval is the parameterization of the prior densities in the diffusion process.

ensures that the decision boundaries established on the source domain are maintained, facilitating the gradual alignment of target features with the support of source ground truth.

The adaptation involves conducting contrastive clustering guided by positive keys generated from the pre-trained diffusion model. To elaborate, the generation of these positive keys follows the sampling procedure outlined in Algorithm 1. We use the notation $(p_0^t \times p_1^t)$ to represent the initial densities in this stage. The prior $p_0^t$ is parameterized by the latent $k$-NNs of a target query $\mathbf{z}^t$. This generation involves a diffusion process $\mathbf{z}_0^t \rightarrow ... \rightarrow \mathbf{z}_{\alpha_t} \rightarrow ... \rightarrow \mathbf{z}_1^t$. With fixed $D$, $\mathbf{z}_1^t$ should correspond to a specific source neighborhood ground truth while reflecting the target-specific latent geometry determined by its $k$-NNs. Appendix C provides the probabilistic formulation of our diffusion model, along with a graphical model that helps in understanding how the diffusion model *stores* and *retrieves* source ground truth throughout the sampling and training processes.

Moreover, to mitigate errors caused by excessive source-related information in the positive key features generated by $D$ due to domain shifts, we propose a feature aggregation technique called *source-informed latent geometry aggregation* (SiLGA). This technique combines diffusion-generated features with features from the query's $k$-NNs for positive key generation. It is inspired by recent research on the impact of latent $k$-NNs on contrastive clustering (Yang et al., 2023). While previous work aligns query predictions with those of its latent $k$-NNs to adjust decision boundaries, SiLGA directly incorporates this latent geometry through feature-level aggregation. For a given target query feature $\mathbf{z}^{t,i}$ with index $i$, we compute its positive key features $\mathbf{z}_{pos}^i$ as follows:

$$\mathbf{z}_{pos}^i := \frac{(\mathbf{z}_1^{t,i} + \mathbb{N}_k(\mathbf{z}^{t,i}))}{k+1}, \tag{4}$$

where $\mathbf{z}_1^{t,i}$ denotes the features generated from $D$ corresponding to the query features $\mathbf{z}^{t,i}$.

### 3.3 Enhancing Contrastive Clustering with Neighborhood Diffusion

In the context of SFDA, the contrastive loss can be formulated in the output probability space (diverging from SSL that operates in the latent space) (Yang et al., 2022; Zhang et al., 2022), due to the fact that the classification task is consistent across the two domains. To maintain brevity and clarity, we use a standard InfoNCE loss (Oord et al., 2018; Chen et al., 2020; He et al., 2020), although various contrastive loss designs exist in contrastive SFDA methods (Yang et al., 2023) for performance

enhancement, and formulate it as follows for each target mini-batch of size $m$:

$$\mathcal{L}_{cls} = -\sum_{i=1}^{m} log \frac{exp(\frac{\sigma[F(\mathbf{z}^{t,i})]^{\top}\sigma[F(\mathbf{z}^{i}_{pos})]}{\tau})}{\sum_{j \neq i} exp(\frac{\sigma[F(\mathbf{z}^{t,i})]^{\top}\sigma[F(\mathbf{z}^{t,j})]}{\tau})}, \quad (5)$$

where $\tau$ denotes the temperature for contrastive logits. We refrain from adding extra techniques like momentum update (He et al., 2020), despite their potential performance benefits, as our focus is to develop a discriminative diffusion model to provide supervision for contrastive clustering.

Intuitively, optimizing a contrastive loss aligns the model's predictions with the clusters represented by the positive keys. Meanwhile, there is a repulsive effect that pushes query predictions away from other samples in the same mini-batch. This interaction affects the generalization error $\epsilon_{\mathbf{T}}(f_t)$, which hinges on how informative these positive keys are regarding the target ground truth. Hence, contrastive clustering in SFDA evaluates how effectively the diffusion process transfers ground truth knowledge between data domains through *source storage* and *target retrieval*.

## 4 EXPERIMENTS

We start by evaluating DND's performance in SFDA (Section 4.1) and subsequently conduct an ablation study to analyze the importance of specific components of our method (Section 4.2).

### 4.1 SOURCE-FREE DOMAIN ADAPTATION

In this section, we evaluate DND's performance on SFDA, specifically its ability to transfer knowledge from the source to the target domain, referred to as *retrieval*. We assess this retrieval capability on the premise that DND can effectively *store* the source ground truth within its samples' neighborhoods. We evaluate the storage ability both on the SFDA benchmarks' source domains (see Appendix D) and on the benchmarks for supervised classification (see Appendix E). Consequently, The accuracy in the target domain serves as an indicator of how well the source knowledge aligns with the target ground truth.

**Datasets**    We conducted our experiments on three widely recognized SFDA benchmarks:
- **Office-31** (Saenko et al., 2010) is a small-scale dataset that consists of 4,652 images spanning 31 object classes collected from three data domains: Amazon (**A**), Webcam (**W**), and DSLR (**D**).
- **Office-Home** (Venkateswara et al., 2017) is a medium-sized dataset containing 15,500 images of 65 classes across four domains: Artistic (**Ar**), Clipart (**Cl**), Product (**Pr**), and Real-World (**Rw**).
- **VisDA-C 2017** (Peng et al., 2017) is a large-scale dataset, used for the 2017 ICCV visual domain adaptation challenge, comprising 280,000 images across 12 classes. The source domain includes synthetic images generated via 3D model rendering, while the target consists of real images.

**Experiment setup**    To ensure replicability, we maintain uniform network architecture and training techniques across all datasets, including ResNet-101 for the encoder $G$, a conditional UNet for the diffusion model $D$, and a two-layer linear classifier $F$. For contrastive clustering, we employ the InfoNCE loss from SimCLR with a logit temperature ($\tau$) set to 0.13 for all datasets. We use an SGD optimizer with a momentum of 0.9 and a mini-batch size of 128, along with a learning rate of $3e^{-3}$ for all datasets. Regarding the number of $k$-NNs, we use three parameters: $k_s^{dif}$ for diffusion model pre-training, $k_t^{dif}$ for diffusion model sampling during target adaptation, and $k_t$ for aggregation with the diffusion-generated features to incorporate target-specific latent geometry. We consistently use 16 diffusion steps for training and sampling in the diffusion model across all experiments, with $k_s^{dif}$, $k_t^{dif}$, and $k_t$ typically set to 15, 15, and 6, respectively.

**Results**    The results for Office-31, Office-Home, and VisDA-C 2017 can be found in Tables 1, 2, and 3, respectively. *ResNet-101* denotes the performance of applying a source pre-trained model directly to the target dataset without any adaptation. Notably, our DND outperforms existing SFDA methods, achieving SOTA results across all three benchmark datasets.

**t-SNE visualization on target feature space**    Figure 3 shows t-SNE visualizations (Van der Maaten & Hinton, 2008) of the target domain's latent feature space after adaptation with the VisDA-C 2017 dataset. We compare scenarios with and without DND-generated features guiding contrastive clustering. Different colors indicate class labels for each target sample. Without DND

Table 1: Comparison of the SFDA methods on *Office-31* (ResNet-50).

| Method | A→D | A→W | D→W | D→A | W→D | W→A | Avg. |
|---|---|---|---|---|---|---|---|
| ResNet-50 (He et al., 2016) | 68.9 | 68.4 | 96.7 | 62.5 | 99.3 | 60.7 | 76.1 |
| SHOT (Liang et al., 2020) | 94.0 | 90.1 | 98.4 | 74.7 | 99.9 | 74.3 | 88.6 |
| 3C-GAN (Li et al., 2020) | 92.7 | 93.7 | 98.5 | 75.3 | 99.8 | **77.8** | 89.6 |
| NRC (Yang et al., 2021a) | 96.0 | 90.8 | 99.0 | 75.3 | 100.0 | 75.0 | 89.4 |
| HCL (Huang et al., 2021) | 94.7 | 92.5 | 98.2 | 75.9 | 100.0 | 77.7 | 89.8 |
| NRC++ (Yang et al., 2023) | 95.9 | 91.2 | **99.1** | 75.5 | 100.0 | 75.0 | 89.5 |
| DND (Ours) | **96.7** | **94.6** | 98.6 | **76.1** | 100.0 | 77.4 | **90.6** |

Table 2: Comparison of the SFDA methods on *Office-Home* (ResNet-50).

| Method | Ar → | | | Cl → | | | Pr → | | | Rw → | | | Avg. |
|---|---|---|---|---|---|---|---|---|---|---|---|---|---|
| | Cl | Pr | Rw | Ar | Pr | Rw | Ar | Cl | Rw | Ar | Cl | Pr | |
| ResNet-50 (He et al., 2016) | 34.9 | 50.0 | 58.0 | 37.4 | 41.9 | 46.2 | 38.5 | 31.2 | 60.4 | 53.9 | 41.2 | 59.9 | 46.1 |
| SHOT (Liang et al., 2020) | 57.1 | 78.1 | 81.5 | 68.0 | 78.2 | 78.1 | 67.4 | 54.9 | 82.2 | 73.3 | 58.8 | 84.3 | 71.8 |
| G-SFDA (Yang et al., 2021b) | 57.9 | 78.6 | 81.0 | 66.7 | 77.2 | 77.2 | 65.6 | 56.0 | 82.2 | 72.0 | 57.8 | 83.4 | 71.3 |
| NRC (Yang et al., 2021a) | 57.7 | 80.3 | 82.0 | 68.1 | 79.8 | 78.6 | 65.3 | 56.4 | 83.0 | 71.0 | 58.6 | 85.6 | 72.2 |
| AaD (Yang et al., 2022) | 59.3 | 79.3 | 82.1 | 68.9 | 79.8 | 79.5 | 67.2 | 57.4 | 83.1 | 72.1 | 58.5 | 85.4 | 72.7 |
| DaC (Zhang et al., 2022) | 59.5 | 79.5 | 81.2 | **69.3** | 78.9 | 79.2 | 67.4 | 56.4 | 82.4 | **74.0** | **61.4** | 84.4 | 72.8 |
| NRC++ (Yang et al., 2023) | 57.8 | **80.4** | 81.6 | 69.0 | 80.3 | 79.5 | 65.6 | 57.0 | 83.2 | 72.3 | 59.6 | 85.7 | 72.5 |
| DND (Ours) | **60.1** | 79.6 | **82.5** | 69.1 | **80.8** | **80.6** | **67.9** | 57.8 | **83.6** | 73.5 | 59.3 | **86.3** | **73.4** |

Table 3: Comparison of the SFDA methods on *VisDA-C 2017* (ResNet-101).

| Method | plane | bcycl | bus | car | horse | knife | mcycl | person | plant | sktbrd | train | truck | Avg. |
|---|---|---|---|---|---|---|---|---|---|---|---|---|---|
| ResNet-101 (He et al., 2016) | 55.1 | 53.3 | 61.9 | 59.1 | 80.6 | 17.9 | 79.7 | 31.2 | 81.0 | 26.5 | 73.5 | 8.5 | 52.4 |
| SHOT (Liang et al., 2020) | 94.3 | 88.5 | 80.1 | 57.3 | 93.1 | 94.9 | 80.7 | 80.3 | 91.5 | 89.1 | 86.3 | 58.2 | 82.9 |
| HCL (Huang et al., 2021) | 93.3 | 85.4 | 80.7 | 68.5 | 91.0 | 88.1 | 86.0 | 78.6 | 86.6 | 88.8 | 80.0 | 74.7 | 83.5 |
| G-SFDA (Yang et al., 2021b) | 96.1 | 88.3 | 85.5 | 74.1 | 97.1 | 95.4 | 89.5 | 79.4 | 95.4 | 92.9 | 89.1 | 42.6 | 85.4 |
| NRC (Yang et al., 2021a) | 96.8 | 91.3 | 82.4 | 62.4 | 96.2 | 95.9 | 86.1 | 80.6 | 94.8 | 94.1 | 90.4 | **59.7** | 85.9 |
| DaC (Zhang et al., 2022) | 96.6 | 86.8 | **86.4** | 78.4 | 96.4 | 96.2 | **93.6** | **83.8** | 96.8 | 95.1 | 89.6 | 50.0 | 87.3 |
| NRC++ (Yang et al., 2023) | 96.8 | 91.9 | 88.2 | 82.8 | 97.1 | 96.2 | 90.0 | 81.1 | 95.2 | 93.8 | 91.1 | 49.6 | 87.8 |
| DND (Ours) | **98.4** | **92.1** | 86.0 | **83.6** | **98.1** | **96.5** | 93.5 | 82.9 | **97.0** | **95.2** | **92.6** | 54.6 | **89.2** |

guidance, the classification model struggles to adapt, leading to significant overlaps among samples from different classes, as shown on the left side of the figure.

In contrast, integrating DND-generated positive keys significantly enhances discriminative clustering accuracy during target adaptation. DND updates the encoder parameters in a way that positions target features inside decision boundaries derived from the source, which results in well-separated clusters. As shown on the right side of the figure, these discriminative clusters align closely with the ground truth labels of target samples, providing robust support for our claim.

This qualitative result serves as a clear demonstration of the effectiveness of our DND in transferring knowledge in the absence of the source data during target adaptation.

## 4.2 ABLATION ANALYSIS OF METHOD COMPONENTS

In the ablation study, we deconstruct our method to understand the roles and impacts of its two core components: the stochastic prior samples and the SiLGA technique used in the positive key generation. We maintain consistency in our experiments by using the VisDA-2017 dataset.

### 4.2.1 PRIOR DENSITY PARAMETERIZATION

Unlike generative models transforming noise densities into meaningful ones, our approach features a more deterministic prior density. While introducing randomness via a Gaussian distribution may bring uncertainty to discriminative feature generation, as mentioned earlier, this randomness can potentially enhance the process by expanding the feature space beyond the encoded space. To in-

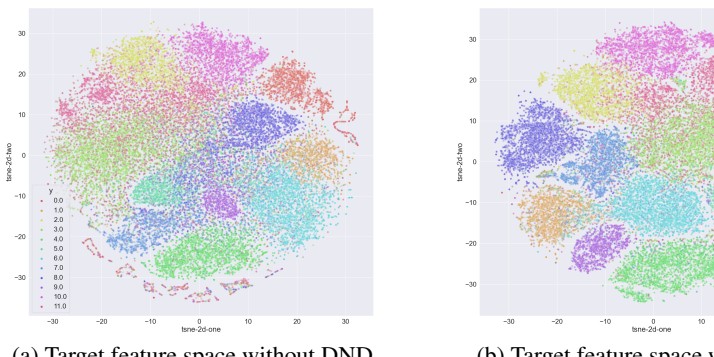

(a) Target feature space without DND.  (b) Target feature space with DND.

Figure 3: (**Best viewed in color.**) The t-SNE visualization of the latent features obtained from the encoded space of the target domain after adaptation has converged, using the *VisDA-C 2017* dataset. Each of the 12 distinct colors corresponds to one of the 12 classes.

vestigate this, we conducted an ablation study where we directly employed the mean of a query's latent $k$-NNs as $\mathbf{z}_0$ for the diffusion process, instead of randomly sampling from a distribution. This setup is labeled as *Ours (without Gaussian Prior)* in Table 7. The results suggest that incorporating randomness via sampling from the prior is significant for diffusion models, irrespective of their generative or discriminative use. For certain scenarios, like $\mathbf{W} \rightarrow \mathbf{A}$, the diffusion-generated features could not operate without such stochastic sampling.

### 4.2.2 FEATURE AGGREGATION FOR POSITIVE KEY GENERATION

To evaluate the effectiveness of our feature aggregation technique *SiLGA*, in generating positive keys, we conduct ablation experiments. The results are presented in Table 4, highlighting the importance of incorporating target $k$-NNs' latent geometry for contrastive clustering. Note that in the table, we have also included a baseline that uses only $k$-NN features for generating positive keys.

Table 4: Experiments to evaluate the contributions of individual DND modules on *VisDA-C 2017*.

| Method | plane | bcycl | bus | car | horse | knife | mcycl | person | plant | sktbrd | train | truck | Avg. |
|---|---|---|---|---|---|---|---|---|---|---|---|---|---|
| ResNet-101 (He et al., 2016) | 55.1 | 53.3 | 61.9 | 59.1 | 80.6 | 17.9 | 79.7 | 31.2 | 81.0 | 26.5 | 73.5 | 8.5 | 52.4 |
| *K*-NN only | 97.5 | 91.1 | 88.6 | 74.6 | 97.4 | 96.2 | 90.8 | 81.6 | 92.6 | 92.8 | 91.5 | 49.9 | 87.1 |
| DND (without Gaussian Prior) | 97.4 | 92.4 | **89.6** | 78.2 | 97.7 | 95.8 | 89.8 | **85.4** | 94.9 | 93.2 | 90.4 | 49.6 | 87.9 |
| DND (without SiLGA) | 97.5 | 92.7 | 89.2 | 78.7 | 97.1 | 95.2 | 86.6 | **85.4** | 93.8 | 92.7 | 92.3 | 50.9 | 87.7 |
| DND (Ours) | **98.4** | **92.1** | 86.0 | **83.6** | **98.1** | **96.5** | **93.5** | 82.9 | **97.0** | **95.2** | **92.6** | **54.6** | **89.2** |

## 5 CONCLUSION

In summary, we delved into the discriminative essence of diffusion generative models, leveraging their potential to solve SFDA problems, supported by the smoothness of latent $k$-NNs. A significant breakthrough in our research was the ability to parameterize prior densities within the diffusion process using the latent geometry informed by $k$-NNs. This advancement allowed us not only to detect domain shifts but also to quantify them explicitly, presenting a robust solution to a foundational issue in SFDA. Through extensive experiments covering both supervised classification and SFDA, we demonstrated the distinct advantages of our DND approach. The incorporation of latent geometry, both in prior density parameterization for diffusion models and feature aggregation for contrastive clustering, has set a new benchmark in SFDA performance.

**Reproducibility Statement** In Section 3, we outline the detailed workflow of our proposed method, supported by a comprehensive explanation of preliminaries and notations in Section 3.1. For a more in-depth understanding of our implementation, we present a probabilistic formulation with detailed graphical models in Appendix C. To ensure reproducibility, we ran experiments with

5 different random seeds, and their results are included in Appendix F. In the experiment setup of both Section 4.1 and Appendix E, we provide implementation details, including the model, optimizer, training iterations, and all hyperparameters used in our method. Additionally, we offer detailed descriptions of the datasets employed in our experiments. For transparency, anonymous code submissions are provided in the supplementary material. All experiments are conducted on public benchmark datasets, and the GitHub repository containing code scripts for our experiments will be made public upon paper acceptance.

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

## A    LATENT GEOMETRY INFORMED BY $k$-NNS

To clarify the intuition behind our method, we frame the SFDA problem within the context of SSL. Here, the challenge arises from the fact that the labeled data is inaccessible while adapting the model to unlabeled data. In this scenario, we can extend the *smoothness assumption*, which is based on the concepts of manifolds and low-density assumptions (Chapelle & Zien, 2005), often employed to address the SSL problem. This assumption implies that similar points are likely to belong to the same cluster. This notion becomes evident when we consider the $k$-Nearest Neighbor ($k$-NN) classifier, which measures the similarity between samples, whether in the input space (Bezdek et al., 1986) or the latent space (Papernot & McDaniel, 2018) in the context of deep learning. In cases where similarities are evaluated in the latent space (*i.e.,* latent $k$-NNs), uncertainties or noise can emerge due to variations in the encoding or decoding processes. This situation highlights the significance of the prediction smoothness of latent $k$-NNs, which can potentially uncover the ground truth of neighboring samples. Expanding upon this insight, we propose extending the concept of latent $k$-NNs and suggest employing it as a guiding principle for the diffusion generative process, which allows us to effectively parameterize its prior densities, enabling the generation of discriminative features for unsupervised clustering. This is, in fact, where the diffusion model takes center stage, generating unseen samples (the samples conform to the underlying data distribution but are absent from the empirical dataset) based on their neighboring data points. Through a stepwise diffusion of ground truth from a labeled sample to its neighbors, the diffusion model effectively revitalizes unseen samples in a specific cluster. This diffusion process seamlessly aligns and, more importantly, harmonizes with the $k$-NN's smoothness assumption. In our work, the diffusion model and latent $k$-NNs collaborate to mitigate cluster noise that may arise during encoding and decoding, or even from the data distribution itself (*e.g.,* domain shifts).

## B    FORMULATING DIFFUSION STEPS WITH STOCHASTIC DIFFERENTIAL EQUATIONS

Brownian motion describes the random fluctuations in the movement of particles within a medium. It is a continuous-time stochastic process that can be viewed as a continuous-time version of a random walk. In the standard Brownian motion, the increments over time follow a normal distribution with a mean of zero and a variance proportional to the time increment. To further describe how the process evolves over time in the absence of random fluctuations, the concept of *drifts* is introduced. These drifts represent deterministic components modeled using an ordinary differential equation (ODE). When combined with Brownian motions, they enable us to formulate an SDE capable of modeling a wide range of stochastic processes that exhibit both deterministic and random behaviors. Thus, a diffusion step can be modeled as the solution to an SDE:

$$d\tilde{\mathbf{z}}_t = g(\tilde{\mathbf{z}}_t, t)dt + \sigma_t(\tilde{\mathbf{z}}_t)dW_t, \tag{6}$$

where $dW_t$ represents the increment of a standard Brownian motion at time $t$. $\tilde{\mathbf{z}}_t$ stands for the transition state at time $t$. if we define the time interval as $t \in [0, 1]$, then $\tilde{\mathbf{z}}_0 \in \mathbb{R}^d$ corresponds to a sample from the prior density, and $\tilde{\mathbf{z}}_1 \in \mathbb{R}^d$ denotes a sample from the target density, *e.g.,* $\tilde{\mathbf{z}}_1 = \mathbf{x}$ when the target is a data density. The drift term $g(\cdot, t)\colon \mathbb{R}^d \to \mathbb{R}^d$ can be parameterized by a neural network (Ho et al., 2020; Luo & Hu, 2021; Wu et al., 2022). The function $\sigma_t(\cdot) : \mathbb{R}^d \to \mathbb{R}^{d \times d}$ is responsible for calculating the covariance matrix of $\tilde{\mathbf{z}}_t$. There are various methods to estimate this covariance matrix, including computing distribution scores (Song et al., 2020) or modeling the conditional distribution of the transition kernel $p(\tilde{\mathbf{z}}_t|\tilde{\mathbf{z}}_0)$ (Song et al., 2021).

## C    DIFFUSION OF GROUND TRUTH WITHIN NEIGHBORHOOD

In this section, we delve into the mechanics of the proposed DND, with a focus on how it diffuses the ground truth knowledge of source samples within their respective neighborhoods during source pre-training and subsequently retrieves this knowledge from target neighborhoods by sampling from the diffusion model during adaptation. As mentioned earlier, the key to knowledge transfer during adaptation without using any source samples is the parameterization of Gaussian priors for both training and sampling from the diffusion model, guiding the diffusion process and enhancing its discriminative essence.

For the sake of clarity, we explain the diffusion generative process in the context of classification tasks like SFDA using probabilistic terms. In Figure 4, we denote an encoding process responsible for transforming input data into their latent features, which follows a distribution $p_\theta(\mathbf{z}|\mathbf{x})$. Moreover, we have a classification process that assigns ground truth labels to the latent features and follows $p_\pi(\mathbf{y}|\mathbf{z})$. In our settings, instead of directly sampling features from the encoded space for classification, we sample features from the diffusion model in the phases after the source representation pre-training. Therefore, during the phases involving the diffusion process, the classification process follows $p_\pi(\mathbf{y}|\mathbf{z}_1)$.

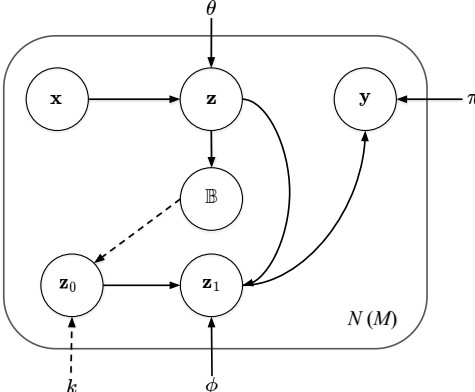

Figure 4: The directed graphical model illustrates the diffusion process for storing and retrieving ground truth within sample neighborhoods. Solid lines represent the direct causal relationships between variables, which include the encoder $p_\theta(\mathbf{z}|\mathbf{x})$, the diffusion generative model $q_\phi(\mathbf{z}_1|\mathbf{z}_0, \mathbf{z})$, and the classifier $p_\pi(\mathbf{y}|\mathbf{z}_1)$. The dashed lines represent the stochastic sampling process from a Gaussian prior $\mathbf{z}_0 \sim q_k(\mathbf{z}_0|\mathbb{B})$. The Gaussian prior is defined over the query's $k$-NN neighborhood.

To parameterize the Gaussian prior, which affects the entire diffusion process like a medium influences the thermodynamics of diffusing substances, we estimate its mean and variance based on the mean and variance of the latent features of the query's $k$-NNs. These neighbors are found within the feature bank $\mathbb{B}$. The latent geometry, determined by these $k$-NNs, serves as the "medium" through which the diffusion process spreads ground truth knowledge around the query sample. Thus, the prior of our diffusion process follows the distribution $q_k(\mathbf{z}_0|\mathbb{B})$, where $\mathbf{z}_0 \sim \mathcal{N}(\mu_k, \sigma_k{}^2)$, and the diffusion process follows $q_\phi(\mathbf{z}_1|\mathbf{z}_0, \mathbf{z})$, where $\mathbf{z}$ is the encoded features from $p_\theta(\mathbf{z}|\mathbf{x})$.

During the source pre-training phase, we define the initial densities $(\mathbf{z}_0^s, \mathbf{z}^s) \sim (q_k(\mathbf{z}_0^s|\mathbb{B}_\mathbf{S}) \times p_\theta(\mathbf{z}^s|\mathbf{x}^s))$, where $\mathbf{z}_0^s \sim \mathcal{N}(\mu_k^s, \sigma_k^{s\,2})$ and $\mathbf{z}^s \sim p_\theta(\mathbf{z}^s|\mathbf{x}^s)$. During this phase, the training objective is to optimize the diffusion model parameters $\phi$ so that it can accurately estimate the expected divergence between $q_k(\mathbf{z}_0^s|\mathbb{B}_\mathbf{S})$ and $p_\theta(\mathbf{z}^s|\mathbf{x}^s)$. The diffusion process progressively transforms the latent densities $(q_k(\mathbf{z}_0^s|\mathbb{B}_\mathbf{S}) \times q_\phi(\mathbf{z}_\alpha^s|\mathbf{z}_0^s, \mathbf{z}^s))$ into the posteriors of the initial densities, which are denoted as $(q_k(\mathbf{z}_0^s|\mathbb{B}_\mathbf{S}) \times p_\theta(\mathbf{z}^s|\mathbf{x}^s))$, with the understanding that $\mathbf{z}_0^s$ is a random variable. Consequently, we can sample $\mathbf{z}_1^s$ from the density $q_\phi(\mathbf{z}_1^s|\mathbf{z}_0^s, \mathbf{z}^s)$ where $\mathbf{z}_1^s \approx \mathbf{z}^s$. Note that, the prior density $q_k(\mathbf{z}_0^s|\mathbb{B}_\mathbf{S})$ is influenced by the latent geometry determined by the query's $k$-NNs. This step-by-step density transformation can be understood as the gradual diffusion (or *storage*) of ground truth knowledge within this neighborhood.

During the target adaptation phase, our goal is to *retrieve* the source ground truth from the target neighborhood. This retrieval process facilitates the update of the encoder parameters $\theta$ during target adaptation. It allows us to make use of the source decision boundaries for classifying unlabeled target samples by integrating the target-specific latent geometry. This latent geometry is determined by the latent $k$-NNs of target queries within the target domain. Thus, the prior for sampling from the diffusion model during target adaptation should follow the distribution $q_k(\mathbf{z}_0^t|\mathbb{B}_\mathbf{T})$, where $\mathbf{z}_0^t \sim \mathcal{N}(\mu_k^t, \sigma_k^{t\,2})$ where $\mu_k^t$ and $\sigma_k^{t\,2}$ denote the mean and variance of the latent $k$-NNs of the target query. As we take more diffusion steps, the sampling process from the diffusion model progressively transforms $\mathbf{z}_0^t$ into $\mathbf{z}_1^t$, which approximates a specific $\mathbf{z}^s$ which shares latent geometry similarities with the current $\mathbf{z}^t$.

Therefore, our diffusion process across the two data domains can be seen as a stochastic mapping from $\mathbf{z}^t$ to $\mathbf{z}^s$. This process involves multiple $\alpha$-deblending and $\alpha$-blending steps, which can be described as follows:

$$(\mathbf{z}_0^t, \mathbf{z}_1^t) \rightarrow \mathbf{z}_{\alpha_0} \rightarrow \cdots \rightarrow \mathbf{z}_{\alpha_t} \cdots \rightarrow \mathbf{z}_{\alpha_T} \rightarrow (\mathbf{z}_0^s, \mathbf{z}_1^s). \tag{7}$$

As the sampling process progresses, the features $\mathbf{z}_{\alpha_t}$ generated by $D$ from $\mathbf{z}_0^t$ become increasingly discriminative as they approach a specific $\mathbf{z}^s$. With enough diffusion steps, $\mathbf{z}_1^t \sim q_\phi(\mathbf{z}_1^t | \mathbf{z}_0^t, \mathbf{z}^t)$ is expected to exhibit similar characteristics to $\mathbf{z}^s \sim p_\theta(\mathbf{z}^s | \mathbf{x}^s)$. In other words, $\mathbf{z}_1^t \approx \mathbf{z}^s$, where the discriminative quality of $\mathbf{z}^s$ is ensured by optimization related to the classification objective in the source domain.

## D  Evaluating DND-Generated Features on the Source Test Sets

To illustrate how well our DND stores source ground truth within the vicinity of a source sample, we utilize features obtained from our pre-trained diffusion model, as outlined in Algorithm 1, for generating classification logits during testing on the source domain. This allows us to evaluate the DND's capacity for *storing* the source ground truth by assessing its classification performance on the source test set, which consists of samples that were not seen during the training.

During the testing phase, we employ *ResNet*-101 to establish the encoded space and the feature bank for neighborhood search. The latent $k$-NNs of a source query are identified based on the encoded features stored in the feature bank. We parameterize the prior density of our DND diffusion process by utilizing the mean and variance of these latent $k$-NNs within a Gaussian distribution. By leveraging the pre-trained *UNet* to estimate the average deviation between the prior density and the density of encoded features, we generate discriminative features through the sampling process of our DND. These features are then employed by the classifier to generate classification logits during testing. These experiments serve to demonstrate the ground truth *storage* capabilities of our DND within the diffusion process, forming the foundation for evaluating *retrieval* during target adaptation. Moreover, the results underscore the aptitude of DND for assessing domain shift even in the absence of source data. The results are reported in Table 5. Surprisingly, the utilization of features sampled from the diffusion process yields better results than the standard testing approach, which motivates us to conduct the follow-up experiments on standard supervised classification benchmarks (Appendix E).

Table 5: Classification accuracy in percentage (%) of using different features for the source domain testing. *ResNet-101 features* indicate the logits for testing are generated from the features sampled from the original encoded space of *ResNet*. The results derived from DND features are highlighted.

| Method | Office-31 | | | Office-Home | | | | VisDA-C 2017 |
|---|---|---|---|---|---|---|---|---|
| | A | D | W | Ar | Cl | Pr | Rw | |
| ResNet-101 features | 91.5 | 100.0 | 98.7 | 81.5 | 81.5 | 93.9 | 85.1 | 99.6 |
| DND features | **96.9** | **100.0** | **100.0** | **96.9** | **90.6** | **100.0** | **96.9** | **100.0** |

## E  Diffusion Process as Latent Augmentation During Testing

This section evaluates the effectiveness of utilizing diffusion-generated features as a form of latent augmentation during the **test phase** of supervised classification. Instead of using samples from the encoded space, we employ features generated by the diffusion model to construct classification logits during the test phase.

**Experimental setup**  During the training of the classification model, we followed the standard settings employed for evaluating the benchmark datasets for supervised classification. To be specific, we utilized the SGD optimizer with a weight decay of 0.0005 and a momentum of 0.9. Our training employed a mini-batch size of 128, and each model underwent 200 epochs of training. For learning rate scheduling, we started with an initial rate of 0.1 and applied the cosine annealing schedule (Loshchilov & Hutter, 2017). Regarding the training of the diffusion model, we maintained the same setup as used in the SFDA experiments, without making any changes to the hyperparameters.

**Datasets**  To evaluate the effectiveness of testing a classification model using diffusion-generated features, we employ three supervised classification benchmarks.

- **CIFAR-10 and CIFAR-100** (Krizhevsky et al., 2009) share 60,000 32×32 color images, with 50,000 for training and 10,000 for testing. In CIFAR-10, there are 10 classes, each with 6,000 images. CIFAR-100 arranges them into 100 classes, with each class comprising 600 images.
- **ImageNet** (Russakovsky et al., 2015), employed in ILSVRC 2012, comprises approximately 1.2 million images spanning 1,000 object categories.

**Results**  Table 6 shows test set classification accuracy in percentage. Our use of small-sized networks is for demonstration, not to achieve state-of-the-art (SOTA) results in supervised classification benchmarks. These findings emphasize that our DND effectively stores ground truth knowledge within the query sample's neighborhood. Features sampled from DND are not only discriminative but also more informative than those from the encoded space. This improvement may be attributed to the parameterization of the prior densities using latent geometry, enabling our DND to generate distinctive features beyond the encoder's capabilities. In essence, during the **test phase** of supervised classification tasks, the diffusion model, trained with fixed classification model parameters, serves as latent-space data augmentation without needing modifications to the classification model parameters.

Table 6: Top-1 test error in percentage (%) on *CIFAR-10*, *CIFAR-100* and *ImageNet*.

| Method | *CIFAR-10* | *CIFAR-100* | *ImageNet* |
|---|---|---|---|
| ResNet-18 | 7.07 | 22.74 | 31.46 |
| ResNet-18 + DND | **6.52 $\pm$ 0.1** | **22.01 $\pm$ 0.2** | **31.06 $\pm$ 0.2** |
| ResNet-50 | 6.35 | 22.23 | 24.68 |
| ResNet-50 + DND | **5.92 $\pm$ 0.2** | **21.95 $\pm$ 0.3** | **24.30 $\pm$ 0.3** |
| VGG-16 | 7.36 | 28.82 | 25.82 |
| VGG-16 + DND | **6.42 $\pm$ 0.1** | **26.58 $\pm$ 0.3** | **25.02 $\pm$ 0.3** |

## F  ROBUSTNESS TO THE SEED FOR INITIALIZATION

To evaluate the stability of our DND model concerning the influence of randomly initialized model parameters during training, we conducted experiments on the *Office-Home* dataset using 5 different initialization seeds. To compare the consistency with other SFDA methods, we replicated the experiments for these methods using the same 5 seeds. To visually capture the variance of the results repeated on different parameter initialization (*i.e.,* different seeds), we present a bar chart with error bars in Figure 5, allowing us to visualize the robustness of various SFDA methods across different random seeds. This analysis provides insights into how reliably each method performs in different initialization scenarios. The results indicate that our DND model, benefiting from the explicit source-informed guidance in contrastive clustering, exhibits greater stability against random initialization compared to other SFDA methods.

## G  ABLATION ANALYSIS OF METHOD COMPONENTS ON OFFICE-31

To evaluate the robustness of our Discriminative Neighborhood Diffusion (DND) on smaller datasets, we replicated the ablation studies outlined in Section 4.2 on the Office-31 dataset. The results, detailed in Table 7, illustrate the impact of each component on the performance within the smaller-scale Office-31 dataset.

## H  HYPERPARAMETER SENSITIVITY ANALYSIS

Our DND requires hyperparameter tuning for effective SFDA. Our primary goal is to demonstrate the discriminative potential of diffusion models, hence our choice of hyperparameters favors a light

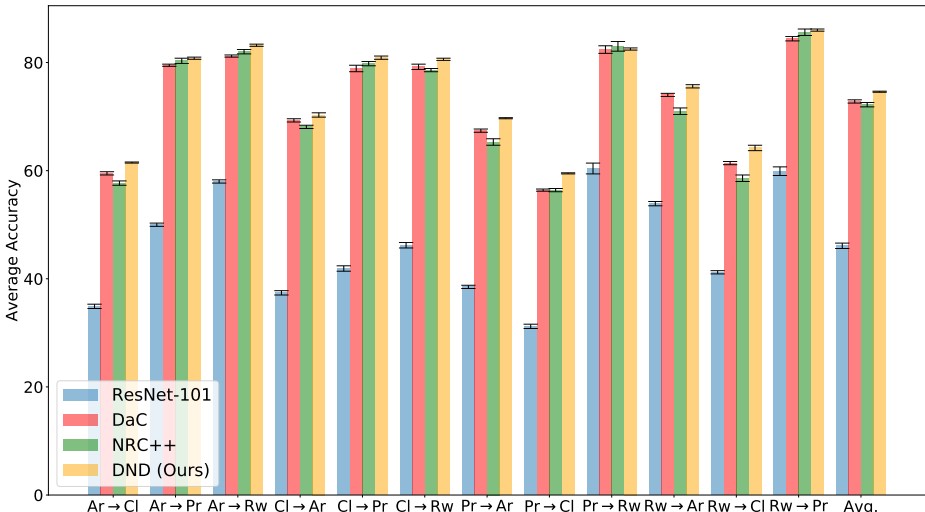

Figure 5: (**Best viewed in color.**) Bar plots with error bars depicting the classification accuracy, measured in percentage (%), across 5 different initialization seeds in the experiments conducted on the *Office-Home* dataset.

Table 7: Experiments to evaluate the contributions of individual DND modules on *Office-31*.

| Method | A→D | A→W | D→W | D→A | W→D | W→A | Avg. |
|---|---|---|---|---|---|---|---|
| ResNet-101 (He et al., 2016) | 70.2 | 69.8 | 94.5 | 52.8 | 97.4 | 62.3 | 74.5 |
| $k$-NN only | 95.0 | 89.6 | 98.2 | 73.4 | 94.3 | 75.1 | 87.6 |
| Ours (without Gaussian Prior) | 93.4 | 84.8 | 98.6 | 55.3 | 98.6 | 59.8 | 81.8 |
| Ours (without SiLGA) | 93.0 | 90.6 | 98.5 | 72.6 | 99.3 | 74.2 | 88.0 |
| Ours (Full) | **96.2** | **94.5** | **98.9** | **76.6** | **100.0** | **76.7** | **90.5** |

model design over maximizing performance. For ease of hyperparameter selection, we have aligned the hyperparameters for training the diffusion model during source pre-training with those used for training the source representation. Furthermore, we have set the number of neighbors for diffusion prior parameterization during target adaptation to match the number of neighbors for neighborhood searching. Meanwhile, we maintain a consistent number of diffusion steps for the sampling process during both source pre-training and target adaptation phases. This approach results in just three distinct hyperparameters for our DND:

- The number of diffusion steps for sampling from the diffusion models.
- The number of neighbors ($k_s$) for diffusion prior parameterization during source pre-training.
- The number of neighbors ($k_t$) for neighborhood searching in the target adaptation phase.

To evaluate the impact of hyperparameter values on the robustness of our DND, we have conducted extensive experiments focusing on the three hyperparameters. In each experiment, we controlled variables as follows: we set $k_s$ to 5 while evaluating $k_t$ and the diffusion steps; $k_t$ was fixed at 5 during the evaluation of $k_s$ and the diffusion steps; and the number of diffusion steps was maintained at 16 when assessing $k_s$ and $k_t$. The results shown in Figure 6 illustrate the robustness of our DND, where our DND outperforms other state-of-the-art SFDA methods across various hyperparameter settings.

## I    ADDITIONAL SOURCE-FREE DOMAIN ADAPTATION EXPERIMENTS USING RESNET-101

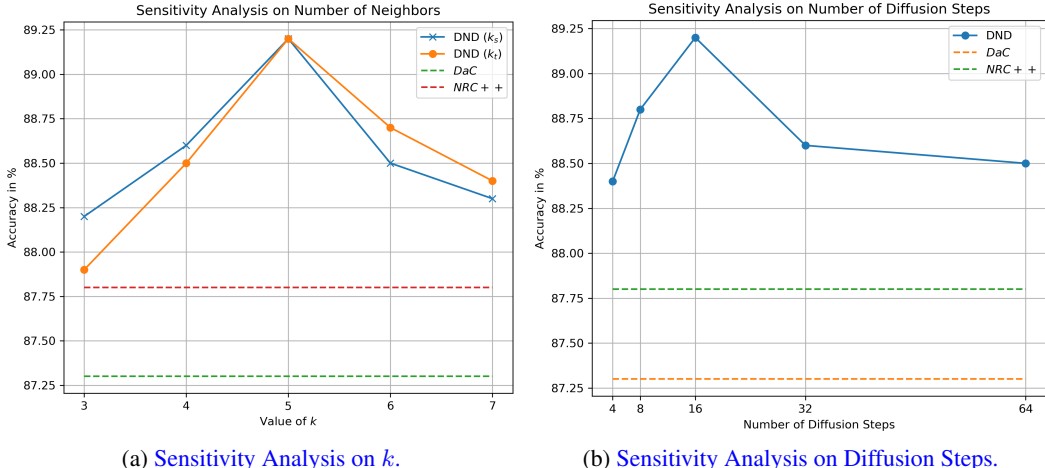

(a) Sensitivity Analysis on $k$.

(b) Sensitivity Analysis on Diffusion Steps.

Figure 6: (**Best viewed in color.**) Hyperparameter Sensitivity Analysis. The results demonstrate the robustness of our DND across a range of different hyperparameter settings.

To validate the robustness of our DND across various backbone networks, we have conducted additional experiments on the Office-31 and Office-Home datasets using the ResNet-101 backbone. The results, presented in Table 8 for Office-31 and Table 13 for Office-Home, illustrate that our DND method effectively addresses the problem of SFDA, regardless of the backbone network used. These findings also confirm that DND consistently surpasses existing SFDA methods in performance, underscoring its adaptability and effectiveness across different network architectures.

Table 8: Comparison of the SFDA methods on *Office-31* with ResNet-101.

| Method | A→D | A→W | D→W | D→A | W→D | W→A | Avg. |
|---|---|---|---|---|---|---|---|
| ResNet-101 (He et al., 2016) | 70.2 | 69.8 | 94.5 | 52.8 | 97.4 | 62.3 | 74.5 |
| SHOT (Liang et al., 2020) | 93.4 | 89.8 | 97.8 | 73.9 | 99.0 | 74.1 | 88.0 |
| 3C-GAN (Li et al., 2020) | 92.0 | 93.4 | 98.2 | 74.8 | 99.4 | 76.2 | 88.5 |
| NRC (Yang et al., 2021a) | 93.8 | 90.8 | 98.1 | 74.2 | 99.6 | 74.6 | 88.5 |
| HCL (Huang et al., 2021) | 94.2 | 92.4 | 98.0 | 75.1 | 99.6 | **76.9** | 89.4 |
| NRC++ (Yang et al., 2023) | 95.1 | 90.5 | 98.2 | 75.3 | 99.6 | 74.2 | 88.8 |
| DND (Ours) | **96.2** | **94.5** | **98.9** | **76.6** | **100.0** | 76.7 | **90.5** |

Table 9: Comparison of the SFDA methods on *Office-Home* (ResNet-101).

| Method | Ar → | | | Cl → | | | Pr → | | | Rw → | | | Avg. |
|---|---|---|---|---|---|---|---|---|---|---|---|---|---|
| | Cl | Pr | Rw | Ar | Pr | Rw | Ar | Cl | Rw | Ar | Cl | Pr | |
| ResNet-101 (He et al., 2016) | 42.6 | 58.8 | 60.2 | 42.6 | 43.4 | 48.1 | 39.2 | 32.9 | 64.2 | 58.5 | 46.8 | 60.2 | 49.8 |
| SHOT (Liang et al., 2020) | 58.4 | 79.6 | 83.2 | 70.1 | 78.2 | 79.4 | 68.2 | 55.4 | 81.6 | 72.8 | 59.2 | 83.2 | 72.4 |
| G-SFDA (Yang et al., 2021b) | 58.9 | 79.2 | 82.4 | 68.3 | 77.9 | 78.6 | 67.8 | 55.8 | 81.4 | 73.6 | 58.5 | 84.4 | 72.2 |
| NRC (Yang et al., 2021a) | 59.4 | 80.2 | 82.6 | 69.2 | 78.6 | 77.9 | 69.2 | 57.2 | 81.8 | 73.6 | 63.4 | 84.3 | 73.1 |
| AaD (Yang et al., 2022) | 60.4 | 80.5 | 82.4 | 69.6 | 79.2 | 78.4 | 68.9 | 58.6 | **83.1** | 73.8 | 63.8 | 85.2 | 73.7 |
| DaC (Zhang et al., 2022) | 60.2 | 80.4 | **83.6** | 68.2 | 79.2 | 80.1 | 67.8 | 58.1 | 81.8 | 74.5 | 62.6 | 85.3 | 73.5 |
| NRC++ (Yang et al., 2023) | 59.7 | 80.3 | 82.6 | 69.5 | 79.6 | 77.6 | 68.5 | 59.2 | 82.2 | 74.2 | 63.5 | 84.9 | 73.5 |
| DND (Ours) | **61.5** | **80.8** | 83.2 | **70.3** | **80.9** | **80.6** | **69.7** | **59.5** | 82.5 | **75.6** | **64.2** | **86.0** | **74.6** |

# J ADDTIONAL EXPERIMENTS ON DOMAIN GENERALIZATION

In adapting our DND to domain generalization, we maintained a consistent approach in the source pre-training stage as per our SFDA settings. This strategy aligns with the widely acknowledged source-free domain generalization framework as discussed in (Cho et al., 2023).

Unlike SFDA, where models might train or fine-tune on target domain data, source-free domain generalization does not involve any training during the testing phase. To adapt our method for this setting, we employed the SiLGA technique to transform target features. This transformation utilized DND-generated source-like features and their adjacent k-nearest neighbors (k-NNs) during target testing, formulated as: $\hat{\mathbf{z}}^{t,i} := \frac{(\mathbf{z}_1^{t,i} + \mathbb{N}_k(\mathbf{z}^{t,i}))}{k+1}$. Here, $\hat{\mathbf{z}}^{t,i}$ represents the transformed target features for classification predictions, $\mathbf{z}_1^{t,i}$ denotes the DND-generated features, and $\mathbb{N}_k(\mathbf{z}^{t,i})$ is the centroid of the latent $k$-NNs of the target features. This process facilitates the application of our method as a data augmentation technique in domain generalization without modifying any model parameters.

In our domain generalization experiments, we concentrated on a single hyperparameter, the number of neighbors ($k$), to investigate latent geometry and define the Gaussian prior for the diffusion process, with $k$ set to 5. The effectiveness of our SiLGA method was evaluated on the large-scale DomainNet dataset, delivering promising results that motivate further exploration of DND in this context. The results on domain generalization experiments are shown in Table 10. Moving forward, we plan to delve into more complex aspects of domain generalization, including methods like averaging batch normalization statistics (Lim et al., 2023) and integrating additional regularization and augmentation techniques.

Table 10: Domain generalization experiments on DomainNet using ResNet-50. The classification accuracies as percentages, obtained by testing on the target domain, are reported.

| Method | DomainNet |
|---|---|
| ZS-CLIP(C) (Radford et al., 2021) | 45.6 |
| CAD (Dubois et al., 2021) | 45.5 |
| ZS-CLIP(PC) (Radford et al., 2021) | 46.3 |
| PromptStyler (Cho et al., 2023) | 49.5 |
| DND (Ours) | **50.8** |

# K  ADDTIONAL EXPERIMENTS ON SINGLE-EPOCH TARGET ADAPTATION

In this section, we investigate the application of our Discriminative Neighborhood Diffusion (DND) model in the context of single-epoch target adaptation, as discussed in (Gao et al., 2023). We aim to evaluate the effectiveness of DND in addressing the challenges associated with single-epoch adaptation in a single-domain setting.

Our method diverges from the work (Gao et al., 2023), where a diffusion model is used as a generative tool, projecting target domain data to source domain data during target inference. This method faces challenges, especially without target labels, as it fails to clearly define the relationship between generated target data and class decision boundaries. We address this by incorporating latent geometry into our diffusion process, which not only provides guidance for establishing this correspondence but also positions our diffusion model as a domain shift corrector during target adaptation. Unlike the generative focus in (Gao et al., 2023), our model transforms target features to align with source-specific characteristics, and we resolve the correspondence issue by parameterizing the diffusion model's prior density through neighborhood searching. This innovative approach marks a significant shift from conventional practices, contributing uniquely to domain adaptation.

Our experimental setup for single-epoch target adaptation focuses on the target inference stage, maintaining our standard source pre-training approach as used in SFDA and aligning with the framework in (Gao et al., 2023). In this setup, adaptation occurs in just one epoch using a contrastive learning objective. The positive key generated by our DND modifies target features to better align with the source feature space. This adaptation aims to align domain-specific features and correct domain shifts, as identified by the diffusion model. Key parameters include setting $k$ to 5 for neighborhood searching, a diffusion step count of 16, and using an SGD optimizer with a learning rate of

$5 \times 10^{-3}$ and a batch size of 128. Initially, experiments were conducted on the ImageNet-C dataset using ResNet-50. The results on the ImageNet-C dataset are provided in Table 11.

Table 11: One-Epoch Target Adaptation Experiments on ImageNet-C with ResNet-50. The classification accuracies as percentages, obtained by testing on the target domain, are reported.

| Method | ImageNet-C |
|---|---|
| MEMO (Radford et al., 2021) | 24.7 |
| DiffPure (Nie et al., 2022) | 16.8 |
| DDA (Gao et al., 2023) | 29.7 |
| DND (Ours) | **32.6** |

## L  RUNTIME ANALYSIS IN TARGET ADAPTATION

In our research, we acknowledge the potential for increased time complexity due to the iterative sampling in our diffusion model. However, our model's primary function is to transform the target feature space, not to generate images, which shapes its design towards efficiency. We have deliberately limited the number of diffusion steps to 16 in all experiments to optimize efficiency. Additionally, our diffusion model does not require estimating a covariance matrix for state transitions, a common step in typical diffusion models, further enhancing efficiency.

In this section, we present a runtime analysis of our DND during target adaptation, comparing it with other contrastive SFDA methods, specifically NRC++ and DaC. To ensure a comprehensive analysis, we chose the large-scale VisDA-2017 dataset for our experiments, which were carried out on a machine equipped with an Nvidia V100 GPU.

Table 12 illustrates that our method and NRC++ have similar one-epoch target adaptation times, while DaC's time complexity is significantly higher. This increased complexity in DaC is due to its adaptive contrastive process and the additional computational requirement to process diverse data sample groups. DaC also includes a self-training phase with pseudo label generation and retraining, contributing to its extended adaptation time. Our experiments show that both DND and NRC++ converge around 10 epochs, while DaC requires about 20 epochs. These findings provide a detailed perspective on the time efficiency of our method in comparison to others, underlining its viability in SFDA settings.

Table 12: Runtime analysis in target adaptation for DND using ResNet-101 on VisDA-2017.

| Method | One Epoch Runtime (s) | Time Required for Convergence (s) |
|---|---|---|
| DaC (Zhang et al., 2022) | 632.8 | 12,656.3 |
| NRC++ (Yang et al., 2023) | 469.2 | 4,692.8 |
| DND (Ours) | 516.3 | 5,163.1 |

## M  ADDTIONAL EXPERIMENTS UNDER PARTIAL-SET DOMAIN ADAPTATION SETTINGS

In this section, we investigate the capability of our DND in managing domain adaptation scenarios with class mismatches. We hypothesize that our DND can effectively generate source-like features, which are informative about specific source ground truths, guided by target-specific latent geometry. Our goal is to determine if our DND can effectively steer the adaptation process to transfer only the knowledge relevant to the target domain's classification tasks in partial-set domain adaptation (PDA). PDA involves adapting models trained on source domains with a broad class range to function effectively in target domains with fewer classes. This adaptation process necessitates the model's ability to discern and concentrate on classes common to both domains, while excluding classes unique to the source domain.

In our approach, DND is used to generate features influenced by the latent geometry of the target domain. During the adaptation phase, DND produces source-like features that are closely aligned in the feature space with the latent geometry of the target query samples. This strategy significantly reduces the risk of generating features corresponding to source-exclusive classes, thus addressing the issue of class mismatches. We validated this claim with experiments on the Office-Home dataset under PDA settings. In the PDA setting, we adhere to the approach outlined in (Cao et al., 2018), selecting the first six classes in alphabetical order as target categories. We then use only the target samples from these six classes for both target adaptation and testing. To accommodate the discrepancy in class ranges, we modified the classifier used in SFDA as SHOT (Liang et al., 2020). This modification included adjusting the number of neurons in the final fully-connected layer of the pretrained source classifier and updating it during target adaptation. To maintain fair comparisons, we applied our method utilizing the existing SHOT codebase, ensuring that no hyperparameters were altered. The results from the Office-Home dataset under PDA settings confirm that our DND can effectively manage class mismatches during adaptation. These experiments underscore the effectiveness of our approach in retrieving source ground truth based on target latent geometry, thereby minimizing negative transfer.

Table 13: Comparison of the partial-set domain adaptation methods on *Office-Home* (ResNet-50).

| Method | Ar → | | | Cl → | | | Pr → | | | Rw → | | | Avg. |
|---|---|---|---|---|---|---|---|---|---|---|---|---|---|
| | Cl | Pr | Rw | Ar | Pr | Rw | Ar | Cl | Rw | Ar | Cl | Pr | |
| ResNet-50 (He et al., 2016) | 46.3 | 67.5 | 75.9 | 59.1 | 59.9 | 62.7 | 58.2 | 41.8 | 74.9 | 67.4 | 48.2 | 74.2 | 61.3 |
| IWAN (Zhang et al., 2018) | 53.9 | 54.5 | 78.1 | 61.3 | 48.0 | 63.3 | 54.2 | 52.0 | 81.3 | 76.5 | 56.8 | 82.9 | 63.6 |
| SAN (Cao et al., 2018) | 44.4 | 68.7 | 74.6 | 67.5 | 65.0 | 77.8 | 59.8 | 44.7 | 80.1 | 72.2 | 50.2 | 78.7 | 65.3 |
| SAFN Xu et al. (2019) | 58.9 | 76.3 | 81.4 | 70.4 | 73.0 | 77.8 | 72.4 | 55.3 | 80.4 | 75.8 | 60.4 | 79.9 | 71.8 |
| SHOT Liang et al. (2020) | 64.8 | 85.2 | 92.7 | 76.3 | **77.6** | 88.8 | 79.7 | 64.3 | 89.5 | **80.6** | 66.4 | 85.8 | 79.3 |
| DND (Ours) | **66.2** | **86.8** | **93.2** | **78.1** | 77.2 | **90.2** | **80.5** | **66.8** | **90.4** | 79.3 | **67.2** | **89.4** | **80.4** |

