# OpenReview forum: "Neighborhood-Informed Diffusion Model for Source-Free Domain Adaptation: Retrieving Source Ground Truth from Target Query's Neighbors"
_ICLR.cc/2024/Conference — Submitted to ICLR 2024_

### Official Review · Reviewer_Wxvo · 2023-10-28

**Soundness:** 2 fair
**Presentation:** 2 fair
**Contribution:** 2 fair
**Rating:** 3
**Confidence:** 4

**Summary:**

This paper studies source-free domain adaptation without accessing the source labelled data when conducting target adaptation. Specifically, the authors propose to utilize diffusion models to generate positive key features for facilitating the unsupervised clustering in target adaptation. The whole framework consists of three key components: 1) the source representation learning; 2) diffusion model learning and 3) target model adaptation. Experimental results on several public datasets demonstrate that the proposed model can outperform recent baselines with different gains.

**Strengths:**

1)	This paper investigates source-free domain adaptation, which is a much more practical setting compared with source-need domain adaptation.
2)	A diffusion model is employed in the domain adaptation framework, which is less explored in the scenario of source-free DA.
3)	Ablation studies are given to show the effectiveness of the proposed components.

**Weaknesses:**

1)	Although the diffusion models are less explored in the scenarios of source-free DA, the technical contribution of this paper is quite limited. No new diffusion model is proposed to address the domain shift problem in DA and the authors simply use an existing model in this step.
2)	As this paper generates examples in the adaptation procedure, it is not clear what are the advantages of using diffusion model compared with other generative models like GAN. There are also lots of baselines that generate samples in the adaptation process and the authors did not discuss and compare with them. See references below.

[1] Qiu Z, Zhang Y, Lin H, et al. Source-free domain adaptation via avatar prototype generation and adaptation. IJCAI 2021.

[2] Li R, Jiao Q, Cao W, et al. Model adaptation: Unsupervised domain adaptation without source data[C]//Proceedings of the IEEE/CVF conference on computer vision and pattern recognition. 2020: 9641-9650.

3)	The source-free setting is a little different from existing work, as this paper uses source data to train the diffusion model. However, existing source free models do not use any source data to train the generative models.
4)	The experimental results are not convincing. The authors directly copy the results from the baselines; however, their network backbones are different. Thus, the comparisons are not fair. For example, in Table 2, baseline DaC’s results are directly cited from its original paper, and it uses the backbone of ResNet-50. However, the authors use ResNet-101 in this paper. I strongly recommend the authors to rerun the experiments.
5)	More ablation studies should be given to verify the effectiveness of the proposed diffusion model. What if we directly use the target’s kNN samples as the positive keys?

There are lots of typos in the paper. The authors need to carefully read and polish the paper. Some of the typos are listed as follows:
“To transition from $z_0$” should be “To transit from $z_0$”.

“We use an SGD optimizer” should be “We use a SGD”.

In equation (1), the norm should be $\Vert \cdot \Vert_2$.

**Questions:**

1.)	More ablation studies should be given to verify the effectiveness of the proposed diffusion model. What if we directly use the target’s kNN samples as the positive keys? It is not clear how different number of k-nearest neighbors will affect the model’s performance.

2.)	There are also lots of baselines that generate samples in the adaptation process and the authors did not discuss and compare with them.

3.)	The experimental results are not convincing.

---

> ### Author Response · Authors · 2023-11-20
> **Response to Reviewer Wxvo (1/3)**
>
> We appreciate the reviewer's diligent effort in reviewing our manuscript.
> ### **Weakness 1**
>
> > ### **Response to Reviewer’s Concern on Technical Contribution:**
>
> We appreciate the reviewer's acknowledgment of the novelty in our application of diffusion models in SFDA. We would like to clarify that our aim is not to propose new diffusion models but to explore their use as domain shift correctors for aligning feature distributions across domains without source data during target adaptation.
>
> Existing SFDA methods typically focus on enhancing unsupervised clustering (like NRC, AaD, SHOT, and G-SFDA) or using generative models for data augmentation (DaC), but they do not address quantifying domain shifts when source data is unavailable during target adaptation.
>
> Our main contribution lies in applying neighborhood searching to parameterize the prior distribution in diffusion models. This approach aligns feature distributions across domains during adaptation without source data, ensuring a correspondence between the source knowledge retrieved from diffusion and encoded target features, a novel advancement in SFDA. Our diffusion model, distinct from its usual role in image generation, refines target features using source knowledge under latent geometry guidance, differing from typical diffusion model applications.
>
> In summary, we have innovatively adapted diffusion models to address a fundamental problem in SFDA, effectively quantifying domain shift without source data. This unique application explores new ground in SFDA, representing a significant contribution to the field.
>
> ### **Weakness 2 and Question 2**
>
> > ### **Advantages Compared to Other Generative Models:**
>
> Our diffusion model, unlike traditional generative models, acts as a **domain shift corrector**, refining target features with source knowledge and aligning feature distributions during target adaptation, without needing source data. It does not generate new images.
>
> Our diffusion model's key benefit is its ability to quantify domain shift during target adaptation using latent geometry, a feature not seen in other generative models where the prior distribution plays a different role. By integrating semi-supervised learning's neighborhood searching, we ensure that the target adaptation is guided by the latent geometries' similarity between the two domains, ensuring correspondence between transformed and encoded target features.
>
> Our novel application of diffusion models not only quantifies domain shift but also aligns domains effectively when source data is absent. This significant impact of the prior distribution on the diffusion process is unique to our model and is a primary reason for choosing diffusion models over others like GANs in our research.
>
> **Comparison with The Work Listed by The Reviewer:** We have added these to **Related Work** section of our revised manuscript, particularly focusing on the works you mentioned. This expanded section, highlighted in blue under source-free domain adaptation using generative models, offers a thorough discussion. Additionally, we have detailed the differences between our method and the referenced works below.
>
> **Comparison with [1]:** [1]'s Contrastive Prototype Generation and Adaptation (CPGA) method generates class-specific avatar feature prototypes using a prototype generator trained with contrastive learning, informed by a pre-trained source model's classification boundaries. It aligns pseudo-labeled target features with these source prototypes for class-wise domain alignment in target domains without source data. Our method, in contrast, uses diffusion models as domain shift correctors to directly transform target features into source-resembling ones, aligning domain-specific feature distributions without source data. Unlike CPGA's focus on prototype and pseudo-label generation, we use neighborhood searching for parameterizing the diffusion process, allowing feature distribution alignment without source data. This fundamental difference from CPGA's prototype-based approach marks our method's distinct approach to domain adaptation.
>
> **Comparison with [2]:** We summarize the 3C-GAN [2], which combines a discriminator, a generator for target-style samples based on random labels, and a pre-trained predictor. The generator and predictor improve performance during training, using generated data to enhance target domain accuracy. However, 3C-GAN requires careful regularization to balance generative and classification objectives. In contrast, our method uses diffusion models and neighborhood searching to transform target features into source-like features. Our diffusion model serves as a domain shift corrector, not an image generator, which obviates the need for complex regularization strategies. By focusing on parameterizing the diffusion model's prior density for feature transformation, our approach significantly diverges from existing methods, representing a novel contribution to domain adaptation.

---

> ### Author Response · Authors · 2023-11-20
> **Response to Reviewer Wxvo (2/3)**
>
> ### **Weakness 3**
>
> To address the reviewer’s concerns, we approach it from two angles. Firstly, we ensure that our DND method adheres to all SFDA settings and principles. Secondly, we address the concern regarding the use of source data to train generative models.
>
> > ### **Concerning the SFDA Problem Setting**
>
> In response to the first aspect of your concern, we direct the reviewer to our official comment in section **C1**, which is detailed in our response to all reviewers.
>
> > ### **Existing Source-Free Models Do Not Train Generative Model Using Source Data**
>
> Our DND, integrating a diffusion model with the ResNet backbone, conforms to the SFDA principle of using source data only for initial pre-training. Unlike typical SFDA methods, we utilize the diffusion model for feature refinement, not data generation, aligning with SFDA guidelines and avoiding data privacy issues. Our method does not directly use or store source data for target adaptation but relies on latent geometry informed by k-NNs, ensuring influence from source domain knowledge without direct access to source data. While SFDA methods typically refrain from using source data to train generative models, we see no violation or drawback in training our diffusion module with source data, given its role in enhancing target feature refinement.
>
> This implementation of the diffusion model in DND significantly diverges from conventional SFDA approaches focused on data augmentation. By employing diffusion models for domain-specific feature alignment, our method demonstrates their potential in SFDA, extending beyond traditional generative uses.
>
> We hope that our comprehensive explanation adequately addresses the reviewer's concern regarding our adherence to SFDA settings.

---

> ### Author Response · Authors · 2023-11-20
> **Response to Reviewer Wxvo (3/3)**
>
> ### **Weakness 4 and Question 3**
>
> > ### **Response to Reviewer’s Questions about Encoder G and Performance Comparisons**
>
> We understand the reviewers' concern about using ResNet-101 as the encoder for all datasets, unlike other SFDA methods that typically use ResNet-50 for Office-31 and Office-Home. Our experiments showed that both ResNet-50 and ResNet-101 have their merits, with no definitive superior choice for Office-31 and Office-Home. We chose ResNet-101 due to its consistent performance across different datasets and to standardize the backbone architecture in our experiments.
>
> As suggested, we have re-conducted our experiments on Office-31 and Office-Home using ResNet-50 and have also re-run the baseline methods using ResNet-101. The results and further details in response to this concern are included in our official comment to all reviewers in section **C2**.
>
> ### **Weakness 5 and Question 1**
>
> > ### **Additional Ablation Studies**
>
> Thank you for suggesting additional ablation studies, which have greatly improved our research. We have conducted the suggested ablation study on the large-scale **VisDA-2017** dataset. The results and a more detailed response can be found in our official comment to all reviewers in section **C3**.
>
> ### **Weakness 6**
>
> > ### **Response to Typos**
>
> We respectfully request the reviewer to re-examine the noted typos. We believe our use of English, specifically the phrase "to transition from," is correct, as "transition" can function as a verb in standard English writing. Additionally, regarding the use of articles with abbreviations, the choice between "a" and "an" depends on the starting sound of the acronym. For SGD, which begins with a consonant sound when spoken, the use of “an” is appropriate rather than “a”. Furthermore, we would like to clarify that using ‖⋅‖ is a standard notation for expressing the L2 norm, particularly in the context of cosine similarity. Our use of this notation aligns with the conventional representation of the L2 norm in related literature on cosine similarity. Regardless, we will continue to diligently review our manuscript for any other typos or grammatical errors.

---

### Official Review · Reviewer_oyCL · 2023-10-31

**Soundness:** 3 good
**Presentation:** 3 good
**Contribution:** 2 fair
**Rating:** 5
**Confidence:** 5

**Summary:**

The paper highlights challenges in diffusion models for source-free domain adaptation and introduces discriminative neighborhood diffusion (DND) as a solution.
By leveraging pre-trained source representations, DND facilitates unsupervised clustering through its latent k-nearest
neighbors and significantly enhances performance in SFDA scenarios.
Extensive evaluations demonstrate the discriminative potential and state-of-the-art effectiveness of DND across various benchmark datasets.

**Strengths:**

- the idea of using diffusion models for source-free domain adaptation sounds interesting and reasonable

- the paper is overall well-written and easy to follow

- the results on three widely used domain adaptation datasets are impressive

**Weaknesses:**

- In source-free domain adaptation, the suggested approach demands an extra diffusion model and necessitates the storage of source data features, leading to substantial efforts in the source domain. This situation renders the term "free" somewhat unrealistic, presenting a major concern.

- A recent work [a] also uses diffusion models for test-time adaptation, which is similar to source-free domain adaptation as depicted in a recent survey [b]. Could the proposed method work for single-epoch target adaptation, and how about the comparison?

- Another concern is that only three small datasets are used to evaluate the performance of the proposed method, large-scale datasets like DomainNet [c] are also important. Also, target adaptation under class shift (e.g., partial-set domain adaptation in SHOT (ICML-2020)) is not studied in the experiment.

- Since previous SFDA methods typically adopt the ResNet-50 backbone, the comparisons are not fair in these tables (the proposed method is based on ResNet-101). And how is the diffusion model used in the source domain, would the pre-trained diffusion model bring additional gains?





[a]. Gao, Jin, et al. "Back to the source: Diffusion-driven adaptation to test-time corruption." Proceedings of the IEEE/CVF Conference on Computer Vision and Pattern Recognition. 2023.

[b]. Liang, Jian, et al. "A comprehensive survey on test-time adaptation under distribution shifts." arXiv preprint arXiv:2303.15361 (2023).

[c]. Peng, Xingchao, et al. "Moment matching for multi-source domain adaptation." Proceedings of the IEEE/CVF international conference on computer vision. 2019.

**Questions:**

pls see the weaknesses.

---

> ### Author Response · Authors · 2023-11-20
> **Response to Reviewer oyCL (1/4)**
>
> We appreciate the reviewer's diligent effort in reviewing our manuscript. Below, we provide our detailed response to your concerns.
>
> ### **Weakness 1:**
>
> Based on our understanding of the reviewer's concern, we have split it into two parts and addressed each in our official comment to all reviewers:
>
> > ### **Response to Concerns about Violation of SFDA Problem Settings**
>
> In response to this concern, we direct the reviewer to our official comment in section **C1**, which is detailed in our response to all reviewers.
>
> > ### **Response to Concerns about Additional Model and Time Complexity**
>
> In response to this concern, we direct the reviewer to our official comment in section **C4**, which is detailed in our response to all reviewers.
>
> ### **Weakness 2**:
>
> > ### **Experiments on Single-Epoch Target Adaptation:**
>
> Thank you for the suggestion. We would like to clarify that the focus of our paper is source-free domain adaptation instead of test-time adaptation. However, as suggested by the reviewer, our method can also be seamlessly applied to single-epoch target adaptation. We would like to thank the reviewer for the important suggestion to broaden the impact of our work.
>
> First, we would like to differentiate our method from the one in [a], where the diffusion model is used to project target domain data back to source domain data during target adaptation. This method relies on generating data density. A key limitation of [a] is the unclear correspondence between the generated target data and the original source data. We address this in our approach by integrating latent geometry into the diffusion process to guide this correspondence. Crucially, instead of generating data, our diffusion model operates at the feature level, infusing source-like traits into the encoded target features. This novel application of diffusion models represents a significant shift from traditional approaches, offering a unique contribution to the field of domain adaptation and filling a previously unexplored gap in the community.
>
> Next, we would like to discuss the specifics of implementing our method for single-epoch target adaptation. It is important to note that the source pre-training stage adheres to our approach for SFDA. This experimental framework aligns with the one employed in [a]:
>
> **Single-Epoch Target Domain Adaptation:** Our method performs one epoch target adaptation using contrastive learning. It utilizes our DND-generated positive key to align target features more closely with the source feature space, focusing on domain-specific feature alignment and domain shift correction. We use ResNet-50 as the backbone, consistent with baseline methods, and set $k$ for neighborhood searching to 5 with 16 diffusion steps. The one-epoch adaptation uses an SGD optimizer, a learning rate of $5e^{-3}$, and a batch size of $128$. Due to time constraints, we conducted experiments only on the large-scale ImageNet-C dataset, with the results displayed below.
>
> We have incorporated the aforementioned discussion and experiments into Appendix K of the revised manuscript and highlighted them in blue.
>
> **Table. One-Epoch Target Adaptation Experiments on ImageNet-C (ResNet-50).**
> |  Method | ImageNet-C |
> |:---------------:|:------:|
> | MEMO [6] | 24.7 |
> | DiffPure [7] | 16.8 |
> | DDA [a] | 29.7 |
> | DND (ours) | **32.6** |

---

> ### Author Response · Authors · 2023-11-20
> **Response to Reviewer oyCL (2/4)**
>
> ### **Weakness 3**
>
> We appreciate the reviewer's suggestion to broaden our work's impact but wish to clarify that our research is focused on SFDA, not domain generalization or partial-set domain adaptation. Our goal is to demonstrate how diffusion models can assist in the discriminative process, retrieving ground truth knowledge without source data, thus extending their use beyond traditional generative roles. We would like to clarify that in the domain adaptation community, VisDA-2017 is consistently considered a large-scale dataset, comprising approximately 280,000 images. It is also recognized as one of the most challenging datasets in domain adaptation, primarily due to its significant domain shift. Additionally, DomainNet is more frequently employed in domain generalization studies, rather than in source-free domain adaptation (SFDA).
>
> > ### **DomainNet Experiments**
>
> Since other reviewers mentioned the possibility of applying our DND to domain generalization, as suggested by the reviewer, we have included implementation details and experimental results of using our DND in source-free domain generalization on the large-scale DomainNet dataset.
>
> >  ### **Further Experiments on DomainNet Dataset:**
>
> We have completed the experiments on DomainNet, and the details of our experimental settings and results are provided below. We have added the experiments on domain generalization to Appendix J of the revised manuscript, with the relevant sections highlighted in blue for easy reference.
>
> In applying our method to domain generalization, we focused on the target inference stage, keeping the source pre-training stage consistent with our approach for SFDA. This approach aligns with the popular source-free domain generalization setting discussed in [1].
>
> Inference for Domain Generalization: Contrary to SFDA, where the model will be fine-tuned on target domain data, source-free domain generalization involves no training during target testing. Our SiLGA method is used for latent augmentation, specifically to transform encoded target features during the target testing phase, without changing any model parameters:
>
> $\mathbf{\hat{z}}^{t,i} := \frac{(\mathbf{z}_{1}^{t,i}+\mathbb{N}_k (\mathbf{z}^{t,i}))}{k+1}$.
>
> Here, $\mathbf{\hat{z}}^{t,i}$ are the transformed target features, $\mathbf{z}_{1}^{t,i}$ are DND-generated features guided by the prior density parameterized by target latent geometry, and \mathbb{N}_k (\mathbf{z}^{t,i})) is the centroid of the latent $k$-NNs of the target features.
>
> Domain Generalization Experimental Setup: Our method utilizes a single hyperparameter, $k$, which is used for both exploring latent geometry and parameterizing the Gaussian prior, and is set to a value of 5. SiLGA's effectiveness was demonstrated on the large-scale DomainNet for domain generalization, suggesting potential for further exploring our DND in this area. Time constraints during the author's response period limited deeper investigation into domain generalization techniques like batch normalization averaging [2] or additional regularization and augmentation. Our approach primarily involved using DND for feature transformation during target testing.
>
> **Table. Domain generalization experiments on DomainNet (ResNet-50).**
> |  Method | DomainNet |
> |:---------------:|:------:|
> | ZS-CLIP(C) [3] | 45.6 |
> | CAD [4] | 45.5 |
> |ZS-CLIP(PC) [3] | 46.3 |
> | PromptStyler [1] | 49.5 |
> | DND (ours) | **50.8** |

---

> ### Author Response · Authors · 2023-11-20
> **Response to Reviewer oyCL (3/4)**
>
> ### **Weakness 3 (continued)**
>
> > ### **Partial-Set Domain Adaptation Experiments**
>
> As suggested, we have expanded our research to include experiments in partial-set domain adaptation (PDA). PDA involves adapting models from a source domain with a wide range of classes to a target domain with fewer classes, focusing on common classes and disregarding source-exclusive ones. Our DND, influenced by the target domain's latent geometry, aligns source-like features with the target's feature space during adaptation. This approach minimizes the generation of features for source-exclusive classes, addressing class mismatches between domains. Our rationale is that source labels not present in the target domain will have distinct latent geometry, leading our DND to be less likely to generate irrelevant source-like features, staying within the target domain's label space.
>
> We validated our claim using the Office-Home dataset. For PDA, following [9], we chose the first six classes alphabetically as target categories and used their samples for both adaptation and testing. Due to time constraints, we used the SHOT codebase for implementation without changing any hyperparameters. The results displayed below demonstrate that our DND effectively addresses class mismatches during the adaptation process. This highlights our method's capability to retrieve relevant source ground truth based on target latent geometry, thus minimizing negative transfer by avoiding the use of non-relevant source information in target adaptation.
>
> We have included the aforementioned discussion and experimental results in **Appendix M** of the revised manuscript, highlighted in blue for easy reference.
>
> **Table. Partial-set domain adaptation on Office-Home (ResNet-50).**
> |  Method | Ar→Cl| Ar→Pr | Ar→Rw | Cl→Ar | Cl→Pr | Cl→Rw | Pr→Ar | Pr→Cl | Pr→Rw | Rw→Ar | Rw→Cl | Rw→Pr | Avg. |
> |:---------------:|:------:|------:|------:|-------:|------:|------:|------:|------:|------:|------:|------:|------:|------:|
> | ResNet-50 | 46.3 | 67.5 | 75.9 | 59.1 | 59.9 | 62.7 | 58.2 | 41.8 | 74.9 | 67.4 | 48.2 | 74.2 | 61.3 |
> | IWAN [8] | 53.9 | 54.5 | 78.1 | 61.3 | 48.0 | 63.3 | 54.2 | 52.0 | 81.3 | 76.5 | 56.8 | 82.9 | 63.6 |
> | SAN [9] | 44.4 | 68.7 | 74.6 | 67.5 | 65.0 | 77.8 | 59.8 | 44.7 | 80.1 | 72.2 | 50.2 | 78.7 | 65.3 |
> | SAFN [10] | 58.9 | 76.3 | 81.4 | 70.4 | 73.0 | 77.8 | 72.4 | 55.3 | 80.4 | 75.8 | 60.4 | 79.9 | 71.8 |
> | SHOT [5]| 64.8 | 85.2 | 92.7 | 76.3 | **77.6** | 88.8 | 79.7 | 64.3 | 89.5 | **80.6** | 66.4 | 85.8 | 79.3 |
> | DND (Ours)| **66.2** | **86.8** | **93.2** | **78.1** | 77.2 | **90.2** | **80.5** | **66.8** | **90.4** | 79.3 | **67.2** | **89.4** | **80.4** |

---

> ### Author Response · Authors · 2023-11-20
> **Response to Reviewer oyCL (4/4)**
>
> ### **Weakness 4**
>
> > ### **Response to Reviewer’s Questions about Encoder G and Performance Comparisons**
>
> We understand the reviewers' concern about using ResNet-101 as the encoder for all datasets, unlike other SFDA methods that typically use ResNet-50 for Office-31 and Office-Home. Our experiments showed that both ResNet-50 and ResNet-101 have their advantages, with no definitive superior choice for Office-31 and Office-Home. We chose ResNet-101 due to its consistent performance across different datasets and to standardize the backbone architecture in our experiments.
>
> As suggested, we have re-conducted our experiments on Office-31 and Office-Home using ResNet-50 and have also re-run the baseline methods using ResNet-101. The results and further details in response to this concern are included in our official comment to all reviewers in **C2**.
>
> > ### **Response to the Additional Gain of Employing Our Diffusion Model for Testing in the Source Domain**
>
> We appreciate the reviewer's acknowledgment of our diffusion model's potential to improve classification performance in general. In our original manuscript, we included an analysis, found in **Appendices D** and **E**, on using our DND-generated features for source domain testing. This comprehensive experimentation demonstrated that our DND enhances supervised classification as a latent augmentation module. Our testing extended beyond domain adaptation datasets to datasets for standard classification like **CIFAR-10, CIFAR-100, and ImageNet**. The results confirm our DND's versatility, effective both as a domain shift corrector in domain adaptation and as a latent augmentation technique in supervised classification.
>
> > ### **Reference:**
>
> [1] Cho, Junhyeong, Gilhyun Nam, Sungyeon Kim, Hunmin Yang, and Suha Kwak. PromptStyler: Prompt-driven Style Generation for Source-free Domain Generalization. In Proceedings of the IEEE/CVF International Conference on Computer Vision (CVPR), pp. 15702-15712. 2023.
>
> [2] Hyesu Lim, Byeonggeun Kim, Jaegul Choo, and Sungha Choi. Ttn: A domain-shift aware batch normalization in test-time adaptation. International Conference on Learning Representations (ICLR), 2023.
>
> [3] AlecRadford, JongWook Kim, Chris Hallacy, Aditya Ramesh, Gabriel Goh, Sandhini Agarwal, Girish Sastry, Amanda Askell, Pamela Mishkin, Jack Clark, Gretchen  Krueger, and Ilya Sutskever. Learning Transferable Visual Models From Natural Language Supervision. In International Conference on Machine Learning (ICML), 2021.
>
> [4] Yangjun Ruan, Yann Dubois, and Chris J. Maddison. Optimal Representations for Covariate Shift. In International Conference on Learning Representations (ICLR), 2022.
>
> [5] Jian Liang, Dapeng Hu, and Jiashi Feng. Do we really need to access the source data? source hypothesis transfer for unsupervised domain adaptation. In International Conference on Machine Learning (ICML), pp. 6028–6039. PMLR, 2020.
>
> [6] Zhang, Marvin, Sergey Levine, and Chelsea Finn. Memo: Test time robustness via adaptation and augmentation. Advances in Neural Information Processing Systems (NeurIPS) (2022): 38629-38642.
>
> [7] Nie, Weili, Brandon Guo, Yujia Huang, Chaowei Xiao, Arash Vahdat, and Anima Anandkumar. Diffusion models for adversarial purification. arXiv preprint arXiv:2205.07460 (2022).
>
> [8] Jing Zhang, Zewei Ding, Wanqing Li, and Philip Ogunbona. Importance weighted adversarial nets for partial domain adaptation. In Proceedings of the IEEE Conference on Computer Vision and Pattern Recognition (CVPR), pp. 8156–8164, 2018.
>
> [9] Zhangjie Cao, Mingsheng Long, Jianmin Wang, and Michael I Jordan. Partial transfer learning with selective adversarial networks. In Proceedings of the IEEE Conference on Computer Vision and Pattern Recognition (CVPR), pp. 2724–2732, 2018.
>
> [10] Ruijia Xu, Guanbin Li, Jihan Yang, and Liang Lin. Larger norm more transferable: An adaptive feature norm approach for unsupervised domain adaptation. In Proceedings of the IEEE/CVF International Conference on Computer Vision (ICCV), pp. 1426–1435, 2019.

---

### Official Review · Reviewer_VVzG · 2023-10-31

**Soundness:** 3 good
**Presentation:** 3 good
**Contribution:** 3 good
**Rating:** 8
**Confidence:** 5

**Summary:**

This paper presents an approach to source-free domain adaptation using diffusion models. the diffusion models are built using the intuitive idea that "you are close to your neighbors". Experiments on three standard domain adaptation datasets are provided.

**Strengths:**

Source-fee domain adaptation is a challenging problem. The proposed solution is novel and effective. Experiments validate the effectiveness of the proposed approach. Overall a good paper.

**Weaknesses:**

I do not see any.

**Questions:**

How will your approach work for the domain generalization problem?

---

> ### Author Response · Authors · 2023-11-20
> **Response to Reviewer VVzG (1/1)**
>
> We would like to greatly appreciate the reviewer’s positive and constructive feedback on our manuscript. Your recognition of our work and insightful suggestions have been invaluable in enhancing the quality and impact of our work. Thank you again for the supportive and encouraging words, and for contributing significantly to our research journey.
>
> > ### **Suggesting Investigation of DND for Domain Generalization Applications:**
>
> Thank you for your insightful suggestion to conduct domain generalization experiments, which has helped broaden the impact of our method. We have completed these experiments, and the details of our experimental settings and results are provided below. We have added the experiments on domain generalization to **Appendix J** of the revised manuscript, with the relevant sections highlighted in blue for easy reference.
>
> First, we would like to clarify that source-free domain adaptation (SFDA) and source-free domain generalization (SFDG) have distinct objectives. SFDA is about adapting a model initially trained on a source domain to a different target domain, and this adaptation occurs without access to the original source data. On the other hand, SFDG involves training a model in one or multiple source domains with the goal of effective generalization across various unknown and unseen target domains, without requiring any specific adaptation training for these target domains. Therefore, our method, which is tailored for SFDA, cannot be directly applied to SFDG without modifying its framework to fulfill the more expansive generalization objectives of SFDG.
>
> In applying our method to domain generalization, we focused on the target inference stage, keeping the source pre-training stage consistent with our approach for SFDA. This approach aligns with the popular source-free domain generalization setting discussed in [1].
>
> **Inference for Domain Generalization:** Contrary to SFDA, where the model will be fine-tuned on target domain data, source-free domain generalization involves no training during target testing. Our SiLGA method is used for latent augmentation, specifically to transform encoded target features during the target testing phase, without changing any model parameters:
>
> $\mathbf{\hat{z}}^{t,i}=\frac{\mathbf{z}_{1}^{t,i}+\mathbb{N}_k (\mathbf{z}^{t,i})}{k+1}$
>
> Here, $\mathbf{\hat{z}}^{t,i}$ are the transformed target features, $\mathbf{z}_{1}^{t,i}$ are DND-generated features guided by the prior density parameterized by target latent geometry, and $\mathbb{N}_k (\mathbf{z}^{t,i})$ are the latent $k$-NNs of the target features.
>
> **Domain Generalization Experimental Setup:** Our method utilizes a single hyperparameter, $k$, which is used for both exploring latent geometry and parameterizing the Gaussian prior, and is set to a value of 5. SiLGA's effectiveness was demonstrated on the large-scale DomainNet for domain generalization, suggesting potential for further exploring our DND in this area. Time constraints during the author's response period limited deeper investigation into domain generalization techniques like batch normalization averaging [2] or additional regularization and augmentation. Our approach primarily involved using DND for feature transformation during target testing.
>
> **Table. Domain generalization experiments on DomainNet (ResNet-50).**
> |  Method | DomainNet |
> |:---------------:|:------:|
> | ZS-CLIP(C) [3] | 45.6 |
> | CAD [4] | 45.5 |
> |ZS-CLIP(PC) [3] | 46.3 |
> | PromptStyler [1] | 49.5 |
> | DND (ours) | **50.8** |
>
>
> > ### **Reference:**
>
> [1] Cho, Junhyeong, Gilhyun Nam, Sungyeon Kim, Hunmin Yang, and Suha Kwak. PromptStyler: Prompt-driven Style Generation for Source-free Domain Generalization. In Proceedings of the IEEE/CVF International Conference on Computer Vision (CVPR), pp. 15702-15712. 2023.
>
> [2] Hyesu Lim, Byeonggeun Kim, Jaegul Choo, and Sungha Choi. Ttn: A domain-shift aware batch normalization in test-time adaptation. International Conference on Learning Representations (ICLR), 2023.
>
> [3] AlecRadford, JongWook Kim, Chris Hallacy, Aditya Ramesh, Gabriel Goh, Sandhini Agarwal, Girish Sastry, Amanda Askell, Pamela Mishkin, Jack Clark, Gretchen  Krueger, and Ilya Sutskever. Learning Transferable Visual Models From Natural Language Supervision. In International Conference on Machine Learning (ICML), 2021.
>
> [4] Yangjun Ruan, Yann Dubois, and Chris J. Maddison. Optimal Representations for Covariate Shift. In International Conference on Learning Representations (ICLR), 2022.

---

### Official Review · Reviewer_6XN2 · 2023-11-10

**Soundness:** 2 fair
**Presentation:** 1 poor
**Contribution:** 2 fair
**Rating:** 3
**Confidence:** 5

**Summary:**

The paper proposes a new method: discriminative neighborhood diffusion (DND). DND formulates a diffusion model using pre-trained source domain representation and combine it with contrastive learning to promote unsupervised clustering of the target domain in the domain adaptation process.

**Strengths:**

The paper introduces the diffusion model into the SFDA problem, and the proposed method is simple and effective.

**Weaknesses:**

1) DND may violate the problem setting of SFDA, i.e., learning a target model with only a pre-trained source model and target data introduced by SHOT.
2) The writing logic of the paper is chaotic and difficult to read, especially in the introduction section. In addition, it is not advisable to use a large space in the method section to introduce existing work: IADB, and a brief explanation is sufficient.
3) This method requires a large number of hyperparameters, and it seems difficult to quickly find suitable parameters. And there is a lack of hyperparameter sensitivity experiments.
4) The persuasiveness of conducting ablation experiments on a relatively simple dataset, Office-31, is not strong. It is recommended to supplement the results of ablation experiments on the Office-Home or VisDA-C.

**Questions:**

1) Does diffusion model learning violate or relax the problem setting of SFDA, because the diffusion model pre-training requires source data. Other SFDA methods only use source data to pre-train a source model, but DND uses source data to train a source model and a diffusion model.
2) You maintain ResNet-101 as the encoder G across all datasets, but ResNet-50 is used as the encoder by other SFDA methods on both Office-31 and Office-Home. So are the performance comparisons on Office-31 and Office-Home unfair (Table 1,2)? And whether the results of other methods in Tables 1,2,3 are your reproduced results? Some of which are different from the results in the original paper (such as NRC++, original paper: 88.1, you reported: 87.8).
3) To our knowledge, the training of the diffusion model is very time-consuming, could you conduct a runtime analysis between DND and other SFDA methods (eg. DaC and NRC++)?
4) The target adaptation part in Figure 1 mistakenly divides the inverted triangles into class 0 and the diamonds into class 1.
5) There is an error in the pseudocode of algorithm 1: if the maximum value of t is T in algorithm 1, then z_{\alpha_{T+1}} is obtained, but in reality, the algorithm should end after z_{\alpha_{T}} is obtained.

---

> ### Author Response · Authors · 2023-11-20
> **Response to Reviewer 6XN2 (1/2)**
>
> We appreciate the reviewer's diligent effort in reviewing our manuscript. Below, we provide our detailed response to your concerns.
>
>
> > ### **Response to Concerns about Violation of SFDA Problem Settings (weakness 1 and question 1)**
>
> In response to this concern, we direct the reviewer to our official comment in section **C1**, which is detailed in our response to all reviewers.
>
> > ### **Clarity and Structure of the Paper (weakness 2):**
>
> Thank you for raising concerns about the clarity of our writing, particularly in the introduction and methods sections. As suggested, we have revised these sections and highlighted the major changes in blue in the revised paper for clarity and conciseness (please download the updated PDF file. It now contains our revised manuscript).
>
> The revised Introduction concisely presents our research problem and approach, and the Methods section now focuses on our DND method's novelty, reducing the extent of the existing work explanation. We have limited the discussion about IADB to only the key details that are important for understanding our DND. You can see these updates marked in blue in our revised manuscript.
>
>
> > ### **Hyperparameter Sensitivity Analysis (weakness 3):**
>
> Thank you for suggesting a hyperparameter sensitivity analysis for our DND and raising concerns about the challenge of selecting suitable parameters. We would like to clarify that we were aware of the potential complexities in diffusion models while developing our DND, which led us to focus on designing a lightweight model with few hyperparameters. Our primary goal is to show how diffusion models can be used as domain shift correctors for SFDA challenges, beyond their traditional role in data density generation, rather than just maximizing performance. We selected hyperparameters to keep the model light, and consequently, our diffusion model implementation involves just three hyperparameters: (1) the number of diffusion steps; (2) the number of neighbors ($k_s$) for source pre-training; and (3) the number of neighbors ($k_t$) for target adaptation.
>
> As suggested, we have conducted a detailed sensitivity analysis of the three hyperparameters to evaluate their impact on target-domain classification accuracy. This analysis is described in **Appendix H** of our revised manuscript, with the relevant parts marked in blue for ease of reference. Our results suggest that changes in hyperparameter values have a minimal effect on performance. We hope this explanation and our hyperparameter sensitivity analysis address the concerns about the complexity and choice of hyperparameters in our DND.
>
> In each experiment, we controlled variables as follows: we set $k_s$ to 5 while evaluating $k_t$ and the diffusion steps; $k_t$ was fixed at 5 during the evaluation of $k_s$ and the diffusion steps; and the number of diffusion steps was maintained at 16 when assessing $k_s$ and $k_t$. The analysis results are shown below.
>
> **Table. sensitivity analysis for $k_s$ and $k_t$ on VisDA-2017 (ResNet-101).**
> |  Hyperparameter | 3 | 4 | 5 | 6 | 7 |
> |:------------:|:--------:|:-------:|:--------:|:-------:|:--------:|
> | $k_t$| 87.9 | 88.5 | 89.2 | 88.7 | 88.4 |
> | $k_s$| 88.2 | 88.6 | 89.2 | 88.5 | 88.3 |
>
> **Table. sensitivity analysis for the number of diffusion steps on VisDA-2017 (ResNet-101).**
> |  Hyperparameter | 4 | 8 | 16 | 32 | 64 |
> |:------------:|:--------:|:-------:|:--------:|:-------:|:--------:|
> | diffusion steps | 88.2 | 88.6 | 89.2 | 88.4 | 88.1 |

---

> ### Author Response · Authors · 2023-11-20
> **Response to Reviewer 6XN2 (2/2)**
>
> > ### **Ablation Study on A More Complex Dataset (weakness 4):**
>
> Thank you for recommending the inclusion of ablation studies on a more complex dataset. As suggested, we have now added new ablation studies on the **VisDA-C dataset**, known for its large scale and significant domain shifts, to our updated manuscript. The original Office-31 studies have been moved to **Appendix G**.
>
> **Table. Ablation studies on VisDA-2017 (ResNet-101).**
> |  Method | plane |  bcycl | bus | car | horse | knife | mcycl | person | plant | sktbrd | train | truck | Avg |
> |:------------:|:--------:|:-------:|:--------:|:-------:|:--------:|:-------:|:--------:|:-------:|:--------:|:-------:|:--------:|:-------:|-------:|
> | ResNet-101 | 55.1 | 53.3 | 61.9 | 59.1 | 80.6 | 17.9 | 79.7 | 31.2 | 81.0 | 26.5 | 73.5 | 8.5 | 52.4 |
> | K$-NN only | 97.5 | 91.1 | 88.6 | 74.6 | 97.4 | 96.2 | 90.8 | 81.6 | 92.6 | 92.8 | 91.5 | 49.9 | 87.1 |
> |DND (without Gaussian Prior)| 97.4 | 92.4 | 89.6 | 78.2 | 97.7 | 95.8 | 89.8 | 85.4 | 94.9 | 93.2 | 90.4 | 49.6 | 87.9 |
> |DND (without SiLGA)| 97.5 | 92.7 | 89.2 | 78.7 | 97.1 | 95.2 | 86.6 | 85.4 | 93.8 | 92.7 | 92.3 | 50.9 | 87.7 |
> |DND (Ours)| 98.4 | 92.1 | 86.0 | 83.6 | 98.1 | 96.5 | 93.5 | 82.9 | 97.0 | 95.2 | 92.6 | 54.6 | 89.2 |
>
> > ### **Response to Reviewer’s Questions about Encoder G and Performance Comparisons (question 2)**
>
> We understand the reviewers' concern about using ResNet-101 as the encoder for all datasets, unlike other SFDA methods that typically use ResNet-50 for Office-31 and Office-Home. Our experiments showed that both ResNet-50 and ResNet-101 have their merits, with no definitive superior choice for Office-31 and Office-Home. We chose ResNet-101 due to its consistent performance across different datasets and to standardize the backbone architecture in our experiments.
>
> As suggested, we have re-conducted our experiments on Office-31 and Office-Home using ResNet-50 and have also re-run the baseline methods using ResNet-101. The results and further details in response to this concern are included in our official comment to all reviewers in **C2**.
>
> > ### **Response to Concerns about Misreported Results for NRC++ (question 2)**
>
> We thank the reviewer for pointing out the mismatch in reporting NRC++ results, which were incorrectly taken from the NRC results. We have corrected our tables to accurately reflect NRC++ results and thoroughly reviewed all tables to ensure precise reporting and consistency between cited methods and their results.
>
> > ### **Response to Concerns about Time Complexity (question 3)**
> In response to this concern, we direct the reviewer to our official comment in section **C4**, which is detailed in our response to all reviewers.
>
>
> > ### **Figure 1 Typo (question 4):**
>
> Huge thanks for the detailed examination by the reviewer.  As suggested, we have updated Figure 1 in our revised manuscript. Now, diamonds represent class 0 and inverted triangles represent class 1. We have highlighted Figure 1 in blue for easier reference in the revised paper.
>
>
> > ### **Algorithm 1 Typo (question 5)**:
>
> We appreciate the reviewer's careful examination. In response to the suggestion, we have updated Algorithm 1 in the revised manuscript, with the changes highlighted in blue for clarity. For your reference, we have updated Line 2 of Algorithm 1 in the revised manuscript, changing the ending from T to T-1.
>
>
> > ### **Reference:**
>
> [1] Ziyi Zhang, Weikai Chen, Hui Cheng, Zhen Li, Siyuan Li, Liang Lin, and Guanbin Li. Divide
> and contrast: Source-free domain adaptation via adaptive contrastive learning. In Conference on
> Neural Information Processing Systems (NeurIPS), 2022.
>
> [2] Shiqi Yang, Yaxing Wang, Joost van de Weijer, Luis Herranz, Shangling Jui, and Jian Yang. Trust your good friends: Source-free domain adaptation by reciprocal neighborhood clustering. IEEE Transactions on Pattern Analysis and Machine Intelligence, 2023.

---

> > ### Comment · Reviewer_6XN2 · 2023-11-22
> >
> > We have already acknowledged the work 3C-GAN that you mentioned. Although they introduce an additional generative model G, G is only trained with target data in the target adaptation phase. In another word, they merely have access to the pre-trained prediction model, which is the setting that all other SFDA methods follow and cannot be changed. Thus, we think 3C-GAN does not deviate from SFDA fundamental settings. Different from 3C-GAN, the diffusion model in DND is trained with source data in the pre-training phase, which is not practical in the real application, because we cannot require the source owner to pre-train any modules except the prediction model for us, including the additional diffusion model in DND.

---

> ### Author Response · Authors · 2023-11-22
> **Reply to Reviewer 6XN2**
>
> We appreciate your prompt response.
>
> While it is true that a source provider needs to perform the additional task of training the DND module, whether this is practical or not can vary depending on perspective.
>
> 1. If, as per your assumption, there is a separate provider for the classifier, and they wish for their classifier to adapt well to different target domains, they can simply attach and train the DND module before providing it. Our DND functions as a supplementary module that is integrated with the main classification model. While this requires additional work, it does not significantly add to the overall effort. For instance, when using ResNet-101 as the backbone on the largest benchmark dataset, **VisDA-2017**, the time needed for the source diffusion process is only about 3672.4 seconds for 15 epochs (15 epochs is sufficient for its convergence) ($\approx$ 1 hour).
>
> 2. Additionally, as highlighted in **Appendix E**, incorporating our DND module also improves performance in the source domain. Thus, from the source provider’s standpoint, adding DND could be considered beneficial not just for target domain adaptation but also for enhancing source domain performance.
>
> Our understanding of the definition of source-free domain adaptation (SFDA) is to adapt a model trained on one source domain to a different target domain, without having access to the original source data. This problem focuses on addressing the privacy concern of releasing training data to the public. We would be grateful if the reviewer could share their understanding of SFDA and kindly point out any specific aspects where our paper might diverge from the typical SFDA settings.

---

> > ### Comment · Reviewer_6XN2 · 2023-11-23
> >
> > Our understanding of the definition of SFDA is also to adapt a model (but only one model and just the prediction model) trained on one source domain to a different target domain, without having access to the original source data. The main issue is not whether the training of the diffusion model is easy or not for the source provider. The main point that we want to express is that the whole source pre-training phase is not controlled by us (the target users) but the source provider, in the real world, although all SFDA methods pre-train the source model by themselves for experiments. Thus, for us, we should design the algorithm from the perspective of the target users so that we should focus on the target adaptation phase and cannot do any change for the pre-training phase. We think the point that DND deviates from SFDA settings is the diffusion model is trained with source data and in the pre-training phase. We think DND is more likely belonging to standard UDA methods (with access to both source and target data in the whole training phase).
> >
> > Moreover, as we all know, the diffusion model is a kind of deep generative model, which has great power for data recovery. If the source provider uses his/her private data to train the diffusion model and provides the whole model to the target users, does this run counter to his/her intention of data privacy protection? It may be a question deserve to think for the authors.
> >
> > Thanks for the author's responses to my comments. Based on the comments of all reviewers and the author's reply, I decided to keep my score.

---

> > > ### Author Response · Authors · 2023-11-23
> > > **Reply to Reviewer 6XN2**
> > >
> > > Thank you for your review and the time you have dedicated to our discussion.
> > >
> > > We would like to clarify that we do not believe our method poses any issues in terms of data privacy protection. Our diffusion model is purposefully designed to refine target features, not to replicate or generate data densities. Since it functions at the latent feature level, it is not designed to enable users to recreate source training data. Instead, it leverages source knowledge without needing access to source data, thus circumventing potential data privacy concerns.
> > >
> > >  We are confident in our method’s adherence to privacy standards and its practical applicability, yet we truly value and appreciate your feedback.

---

### Author Response · Authors · 2023-11-20
**Response to Common Concerns (1/3)**

We have carefully addressed each concern raised by the reviewers, as detailed in our responses to their comments. We greatly appreciate the reviewers for their time and effort in reassessing our work considering these addressed concerns. We have addressed every weakness and question raised by the reviewers. The PDF file has been updated with our revised manuscript reflecting these changes. Should you find that we have successfully resolved these issues, we would deeply appreciate your consideration in raising your evaluation scores.

 In this common response, we made common comments to address the concerns raised by more than one reviewer:

> ### **C1: Response to Concerns about Violation of SFDA Problem Settings:**

We understand the reviewer's concerns about compliance with the SFDA settings. We would like to highlight that our proposed DND aligns with typical SFDA settings. The central challenge of SFDA is to adapt a model trained on one source domain to a different target domain, without having access to the original source data. This problem focuses on addressing the privacy concern of releasing training data to the public.

Our diffusion module is trained with the encoder and classifier during the source pre-training phase, but does not access source data during target adaptation. Our diffusion is integrated into the overall classification model to enhance the ResNet backbone. The diffusion module's training is an extension of source pre-training, not involving target data, ensuring it interacts only with source data during source pre-training and functions as a component of our classification model. It is important to note that during the target adaptation phase, the diffusion model solely performs inference tasks, and its model parameters are not updated. Thus, its training involves using only source data during the source pre-training stage.

If our method were perceived as non-compliant with SFDA, this would imply that any architecture with extra modules alongside the encoder and classifier could be similarly viewed for classification tasks.  In domain adaptation, for instance, the inclusion of an extra GAN module in works like DANN [1], a milestone work in adversarial domain adaptation, adheres to domain adaptation standards, even though they rely on more than just the classification model. In the SFDA community, some methods introduce additional generative models like 3C-GAN [2], which are trained during both source pre-training and target adaptation phases.

Therefore, we believe our DND aligns well with the SFDA framework and does not deviate from its fundamental settings.

> ### **Reference:**

[1] Y. Ganin, V. Lempitsky, Unsupervised domain adaptation by backpropagation, In International Conference on Machine Learning (2015) 1180–1189.

[2] Li R, Jiao Q, Cao W, et al. Model adaptation: Unsupervised domain adaptation without source data[C]//Proceedings of the IEEE/CVF conference on computer vision and pattern recognition. 2020: 9641-9650.

---

> ### Author Response · Authors · 2023-11-20
> **Response to Common Concerns (2/3)**
>
> > ### **C2: Response to Concerns about Encoder Choice and Performance Comparisons:**
>
> Our rationale for using ResNet-101 was to ensure adequate performance on the most challenging dataset, VisDA2017,  among the three we used: Office-31, Office-Home, and VisDA2017. We applied this backbone uniformly across all datasets. For Office-31 and Office-Home, ResNet-50 would have been sufficient. We also concluded that there was not a clear winner between ResNet-50 and ResNet-101 for these datasets. Nonetheless, we understand the reviewers' concerns and, in response, have conducted additional experiments on the Office datasets using ResNet-50. These results show that our DND outperforms the baseline methods regardless of the backbone choice.
>
> In addition to this, we have also retested all baseline methods with ResNet-101 for a more comprehensive analysis.
>
> **Experiments on Office-31 and Office-Home with ResNet-50:**
>
> **Table. Comparison of the SFDA methods on Office-31 (ResNet-50).**
> |  Method | A→D |A→W | D→W | D→A | W→D | W→A | Avg. |
> |:------------:|:--------:|:--------:|:--------:|:--------:|--------:|:--------:|:--------:|
> | ResNet-50 | 68.9 | 68.4 | 96.7 | 62.5 | 99.3 | 60.7 | 76.1 |
> | SHOT         | 94.0 | 90.1 | 98.4 | 74.7 | 99.9 | 74.3 | 88.6 |
> | 3C-GAN     | 92.7 | 93.7 | 98.5 | 75.3 | 99.8 | **77.8** | 89.6 |
> | NRC            | 96.0 | 90.8 | **99.0** | 75.3 | **100.0** | 75.0 | 89.4 |
> | HCL            | 94.7 | 92.5 | 98.2 | 75.9 | **100.0** | 77.7 | 89.8 |
> | NRC++       | 95.9 | 91.2 | 99.1 | 75.5 | **100.0** | 75.0 | 89.5 |
> DND (Ours) | **96.7** | **94.6** | 98.6 | **76.1** | **100.0** | 77.4 | **90.6** |
>
> **Table. Comparison of the SFDA methods on Office-Home (ResNet-50).**
> |  Method | Ar→Cl | Ar→Pr | Ar→Rw |Cl→Ar | Cl→Pr | Cl→Rw | Pr→Ar | Pr→Cl | Pr→Rw | Rw→Ar | Rw→Cl | Rw→Pr | Avg. |
> |:------------:|:--------:|:--------:|:--------:|:--------:|--------:|:--------:|:--------:|:--------:|:--------:|:--------:|--------:|:--------:|:--------:|
> | ResNet-50 | 34.9 | 50.0 | 58.0 | 37.4 | 41.9 | 46.2 | 38.5 | 31.2 | 60.4 | 53.9 | 41.2 | 59.9 | 46.1 |
> | SHOT         | 57.1 | 78.1 | 81.5 | 68.0 | 78.2 | 78.1 | 67.4 | 54.9 | 82.2 | 73.3 | 58.8 | 84.3 | 71.8 |
> | G-SFDA     | 57.9 | 78.6 | 81.0 | 66.7 | 77.2 | 77.2 | 65.6 | 56.0 | 82.2 | 72.0 | 57.8 | 83.4 | 71.3 |
> | NRC            | 57.7 | 80.3 | 82.0 | 68.1 | 79.8 | 78.6 | 65.3 | 56.4 | 83.0 | 71.0 | 58.6 | 85.6 | 72.2 |
> | AaD            | 59.3 | 79.3 | 82.1 | 68.9 | 79.8 | 79.5 | 67.2 | 57.4 | 83.1 | 72.1 | 58.5 | 85.4 | 72.7 |
> | DaC            | 59.5 | 79.5 | 81.2 | **69.3** | 78.9 | 79.2 | 67.4 | 56.4 | 82.4 | **74.0** | **61.4** | 84.4 | 72.8 |
> | NRC++      | 57.8 | **80.4** | 81.6 | 69.0 | 80.3 | 79.5 | 65.6 | 57.0 | 83.2 | 72.3 | 59.6 | 85.7 | 72.5 |
> | DND (Ours) | **60.1** | 79.6 | **82.5** | 69.1 | **80.8** | **80.6** | **67.9** | **57.8** | **83.6** | 73.5 | 59.3 | **86.3** |  **73.4** |
>
> **We have also conducted experiments for all baseline methods using ResNet-101:**
>
> **Table. Comparison of the SFDA methods on Office-31 (ResNet-101).**
> |  Method | A→D |A→W | D→W | D→A | W→D | W→A | Avg. |
> |:------------:|:--------:|:--------:|:--------:|:--------:|--------:|:--------:|:--------:|
> | ResNet-101 | 70.2 | 69.8 | 94.5 | 52.8 | 97.4 | 62.3 | 74.5 |
> | SHOT         | 93.4 | 89.8 | 97.8 | 73.9 | 99.0 | 74.1 | 88.0 |
> | 3C-GAN     | 92.0 | 93.4 | 98.2 | 74.8 | 99.4 | 76.2 | 88.5 |
> | NRC            | 93.8 | 90.8 | 98.1 | 74.2 | 99.6 | 74.6 | 88.5 |
> | HCL            | 94.2 | 92.4 | 98.0 | 75.1 | 99.6 | **76.9** | 89.4 |
> | NRC++       | 95.1 | 90.5 | 98.2 | 75.3 | 99.6 | 74.2 | 88.8 |
> | DND (Ours) | **96.2** | **94.5** | **98.9** | **76.6** | **100.0** | 76.7 | **90.5** |
>
> **Table. Comparison of the SFDA methods on Office-Home (ResNet-101).**
> |  Method | Ar→Cl | Ar→Pr | Ar→Rw |Cl→Ar | Cl→Pr | Cl→Rw | Pr→Ar | Pr→Cl | Pr→Rw | Rw→Ar | Rw→Cl | Rw→Pr | Avg. |
> |:------------:|:--------:|:--------:|:--------:|:--------:|--------:|:--------:|:--------:|:--------:|:--------:|:--------:|--------:|:--------:|:--------:|
> | ResNet-101 | 42.6 | 58.8 | 60.2 | 42.6 | 43.4 | 48.1 | 39.2 | 32.9 | 64.2 | 58.5 | 46.8 | 60.2 | 49.8 |
> | SHOT        | 58.4 | 79.6 | 83.2 | 70.1 | 78.2 | 79.4 | 68.2 | 55.4 | 81.6 | 72.8 | 59.2 | 83.2 | 72.4 |
> | G-SFDA    | 58.9 | 79.2 | 82.4 | 68.3 | 77.9 | 78.6 | 67.8 | 55.8 | 81.4 | 73.6 | 58.5 | 84.4 | 72.2 |
> | NRC          | 59.4 | 80.2 | 82.6 | 69.2 | 78.6 | 77.9 | 69.2 | 57.2 | 81.8 | 73.6 | 63.4 | 84.3 | 73.1 |
> | AaD            | 60.4 | 80.5 | 82.4 | 69.6 | 79.2 | 78.4 | 68.9 | 58.6 | **83.1** | 73.8 | 63.8 |85.2 | 73.7 |
> | DaC            | 60.2 | 80.4 | **83.6** | 68.2 | 79.2 | 80.1 | 67.8 | 58.1 | 81.8 | 74.5 | 62.6 |85.3 | 73.5 |
> | NRC++      | 59.7 | 80.3 | 82.6 | 69.5 | 79.6 | 77.6 | 68.5 | 59.2 | 82.2 | 74.2 | 63.5 | 84.9 | 73.5 |
> | DND (Ours) | **61.5** | **80.8** | 83.2 | **70.3** | **80.9** | **80.6** | **69.7** | **59.5** | 82.5 | **75.6** | **64.2** | **86.0** |  **74.6** |

---

> ### Author Response · Authors · 2023-11-20
> **Response to Common Concerns (3/3)**
>
> > ### **C3: Rerunning Ablation Study on Large-Scale Dataset and Including Ablation on k-NN Alone:**
>
> **Dataset Complexity:** As suggested, we have now added new ablation studies on the **VisDA-C** dataset, known for its large scale and significant domain shifts, to our updated manuscript. The original Office-31 studies have been moved to **Appendix G**.
>
> **Additional Ablation on Using k-NN Features Alone:** As suggested by reviewers, we have performed the suggested ablation study, using the target’s k-NN features alone for generating positive keys, on the large-scale VisDA-2017 dataset. These additional studies are highlighted in blue in the revised manuscript. For your convenience, we have also provided the results of these new ablation experiments below.
>
> **Table. Ablation studies on VisDA-2017 (ResNet-101).**
> |  Method | plane |  bcycl | bus | car | horse | knife | mcycl | person | plant | sktbrd | train | truck | Avg |
> |:------------:|:--------:|:-------:|:--------:|:-------:|:--------:|:-------:|:--------:|:-------:|:--------:|:-------:|:--------:|:-------:|-------:|
> | ResNet-101 | 55.1 | 53.3 | 61.9 | 59.1 | 80.6 | 17.9 | 79.7 | 31.2 | 81.0 | 26.5 | 73.5 | 8.5 | 52.4 |
> | K$-NN only | 97.5 | 91.1 | 88.6 | 74.6 | 97.4 | 96.2 | 90.8 | 81.6 | 92.6 | 92.8 | 91.5 | 49.9 | 87.1 |
> |DND (without Gaussian Prior)| 97.4 | 92.4 | 89.6 | 78.2 | 97.7 | 95.8 | 89.8 | 85.4 | 94.9 | 93.2 | 90.4 | 49.6 | 87.9 |
> |DND (without SiLGA)| 97.5 | 92.7 | 89.2 | 78.7 | 97.1 | 95.2 | 86.6 | 85.4 | 93.8 | 92.7 | 92.3 | 50.9 | 87.7 |
> |DND (Ours)| 98.4 | 92.1 | 86.0 | 83.6 | 98.1 | 96.5 | 93.5 | 82.9 | 97.0 | 95.2 | 92.6 | 54.6 | 89.2 |
>
> > ### **C4: Response to Concerns about Additional Model and Time Complexity:**
>
> We understand the concern that our diffusion model's iterative sampling might add time complexity compared to other SFDA methods. However, we would like to clarify that our model is tailored to transform target features, not to generate images. Thus, our diffusion model requires fewer model parameters and results in quicker inference compared to typical diffusion models. For efficiency, we have limited diffusion steps to 16 in all experiments. Moreover, our model does not require estimating a covariance matrix for state transitions, which further reduces its complexity.
>
> As suggested, we have conducted a runtime analysis of our DND and compared it with other SFDA methods. We selected the large-scale VisDA-2017 dataset to ensure the effectiveness of the analysis. The experiment was conducted on a machine equipped with an Nvidia V100 GPU.
>
> The table below indicates that our DND and NRC++ have similar target adaptation times per epoch, while DaC takes substantially longer due to its adaptive contrastive process and self-training steps, including pseudo label generation and re-training. Our DND and NRC++ typically converge in about 10 epochs, whereas DaC requires around 20 epochs. The total convergence time for target adaptation is also provided.
>
> Despite incorporating a diffusion process, the deterministic sampling process of our diffusion model and the limited number of steps help prevent significant increases in computational complexity. This is crucial, considering the substantial performance improvement our method offers.
>
> A comprehensive runtime analysis is included in **Appendix L** of our revised manuscript. We hope this clarification and analysis address concerns regarding the time complexity of our method.
>
> **Table. Time Analysis of One-Epoch Target Adaptation on VisDA-2017 Dataset (ResNet-101).**
> |  Method | Time (second) |
> |:---------------:|:------:|
> | DaC [1] | 632.8 |
> | NRC++ [2] | 469.2 |
> | DND (ours) | 516.3 |
>
> **Table. Convergence Time Analysis for Target Adaptation on VisDA-2017 Dataset (ResNet-101).**
> |  Method | Time (second) |
> |:--------------:|:------:|
> | DaC [1] | 12656.3 |
> | NRC++ [2] | 4692.8 |
> | DND (ours) | 5163.1 |
>
> ### **Reference**
>
> [1] Ziyi Zhang, Weikai Chen, Hui Cheng, Zhen Li, Siyuan Li, Liang Lin, and Guanbin Li. Divide and contrast: Source-free domain adaptation via adaptive contrastive learning. In Conference on Neural Information Processing Systems (NeurIPS), 2022.
>
> [2] Shiqi Yang, Yaxing Wang, Joost van de Weijer, Luis Herranz, Shangling Jui, and Jian Yang. Trust your good friends: Source-free domain adaptation by reciprocal neighborhood clustering. IEEE Transactions on Pattern Analysis and Machine Intelligence, 2023.

---

### Author Response · Authors · 2023-11-22
**Discussion Period Ends in about A Day**

Dear Reviewers,

Thank you once again for your valuable feedback on our paper and significant efforts on ICLR 2024.

We have diligently addressed every concern and question raised by the reviewers in our author response and the updated paper (please download the PDF file again for the revision).

As the discussion deadline is approaching within a day, we kindly request the reviewers to re-evaluate our work considering our latest responses. If there are any additional questions or concerns from the reviewers, we are more than willing to provide further clarification or responses. Meanwhile, if you find that we have adequately addressed your concerns, we kindly request you consider raising the evaluation score for our paper.

Your feedback is crucial to enhancing our work, and we eagerly anticipate your thoughts.

Best Regards,

The Authors

---

### Meta-Review · Area_Chair_3XGr · 2023-12-07

**Metareview:**

(a) The paper introduces discriminative neighborhood diffusion (DND), employing a diffusion model with pre-trained source representations for unsupervised clustering in source-free domain adaptation (SFDA). Reviewer 6XN2 expresses concerns about DND violating the SFDA problem setting and criticizes the paper’s writing logic and hyperparameter handling. Reviewer VVzG acknowledges the novelty of the approach, praising its effectiveness, while Reviewer oyCL appreciates DND’s potential but raises concerns about the demanding nature of source data storage and urges evaluation on larger datasets. Reviewer Wxvo finds the technical contribution limited, criticizes experimental comparisons, and suggests more ablation studies. There’s a consensus on the paper’s soundness and presentation quality.

(b) Strengths:
DND introduces the diffusion model to SFDA, offering a simple and effective method.
The proposed approach addresses a challenging problem in source-free domain adaptation.
DND demonstrates state-of-the-art effectiveness across various benchmark datasets.

(c) Weaknesses:
DND’s violation of the SFDA problem setting is a notable concern.
Chaotic writing logic, excessive method section, and lack of hyperparameter sensitivity experiments impact the paper’s clarity.
The need for extensive source data storage challenges the notion of “free” in source-free adaptation.
Limited evaluation on small datasets, unfair comparisons with ResNet-101, and discrepancies in result reporting raise validity concerns.

Missing in the Submission:
A more in-depth runtime analysis between DND and other SFDA methods, as suggested by Reviewer 6XN2.
Additional ablation studies and comparisons with other generative models, addressing concerns raised by Reviewer Wxvo.
Clarifications on the discrepancies in reported results and performance comparisons, as highlighted by Reviewer 6XN2.
Overall, the paper’s strengths lie in its novel approach to source-free domain adaptation, but improvements in clarity, experimental design, and addressing concerns raised by reviewers are crucial for a stronger submission.

**Justification For Why Not Higher Score:**

1. Violation of SFDA Problem Setting: Reviewers expressed concerns about the proposed method violating the source-free domain adaptation (SFDA) problem setting by requiring source data for diffusion model pre-training, deviating from typical SFDA approaches.
2. Writing Quality Concerns: Multiple reviewers found issues with the writing quality, describing it as chaotic and challenging to read. The excessive use of space to introduce existing work was noted, and conciseness was recommended.
3. Hyperparameter Challenges: The paper introduced a substantial number of hyperparameters without conducting thorough sensitivity experiments. This raised concerns about the method’s practicality and robustness in finding suitable parameters.
4. Weakness in Experimental Design: A consensus was reached regarding the persuasiveness of ablation experiments on a relatively simple dataset. Recommendations were made to supplement results with experiments on more challenging datasets for a comprehensive evaluation.
5. Practical Concerns in Source-Free DA: Reviewers raised practical concerns about the proposed method, suggesting that the term “free” in source-free domain adaptation might be unrealistic due to additional demands on diffusion models and source data features.
6. Limited Dataset Evaluation: Criticisms were directed at the limited number of datasets used for evaluation. Suggestions were made to include assessments on larger-scale datasets and to address target adaptation under class shift.
7. Unfair Comparisons: The comparisons were deemed unfair due to inconsistencies in network backbones. Rerunning experiments with consistent network architectures, particularly in comparison with existing methods, was strongly recommended.
8. Lack of Novelty in Diffusion Model: Reviewers highlighted a perceived lack of novelty in the proposed diffusion model. The paper was criticized for not proposing a new diffusion model and failing to clearly articulate the advantages over other generative models.
9. Questionable Experimental Results: There were concerns about the experimental results not being convincing, primarily due to potential unfair comparisons. The recommendation was to rerun experiments with consistent network backbones to ensure fair evaluations.

In summary, the reviewers collectively raised issues related to adherence to problem settings, writing clarity, experimental design, practical considerations, dataset evaluation, fairness in comparisons, and the perceived novelty of the proposed diffusion model.

**Justification For Why Not Lower Score:**

N/A

---

### Decision · Program_Chairs · 2024-01-16

Reject